# Phenotypic and Molecular-Phylogenetic Analyses Reveal Distinct Features of Crown Gall-Associated *Xanthomonas* Strains

Hamzeh Mafakheri,[a,b] S. Mohsen Taghavi,[a] Sadegh Zarei,[a,b] [ID]Touraj Rahimi,[c] Mohammad Sadegh Hasannezhad,[d] [ID]Perrine Portier,[e] [ID]Marion Fischer-Le Saux,[e] [ID]Ivica Dimkić,[f] [ID]Ralf Koebnik,[g] [ID]Nemanja Kuzmanović,[h*] [ID]Ebrahim Osdaghi[b]

[a]Department of Plant Protection, School of Agriculture, Shiraz University, Shiraz, Iran
[b]Department of Plant Protection, College of Agriculture, University of Tehran, Karaj, Iran
[c]Department Plant Production and Genetics, Agriculture Faculty, Shahr-e-Qods Branch, Islamic Azad University, Tehran, Iran
[d]University of Applied Science and Technology, Sadra, Iran
[e]Université Angers, Institut Agro, INRAE, IRHS, SFR QUASAV, CIRM-CFBP, Angers, France
[f]Department of Biochemistry and Molecular Biology, University of Belgrade Faculty of Biology, Belgrade, Serbia
[g]Plant Health Institute of Montpellier, University of Montpellier, CIRAD, INRAE, Institut Agro, IRD, Montpellier, France
[h]Julius Kühn-Institut, Federal Research Centre for Cultivated Plants, Institute for Epidemiology and Pathogen Diagnostics, Braunschweig, Germany

**ABSTRACT** In summer 2019, widespread occurrence of crown gall disease caused by *Agrobacterium* spp. was observed on commercially grown ornamental plants in southern Iran. Beside agrobacteria, pale yellow-pigmented Gram-negative strains resembling the members of *Xanthomonas* were also associated with crown gall tissues on weeping fig (*Ficus benjamina*) and *Amaranthus* sp. plants. The purpose of the present study was to characterize the crown gall-associated *Xanthomonas* strains using plant inoculation assays, molecular-phylogenetic analyses, and comparative genomics approaches. Pathogenicity tests showed that the *Xanthomonas* strains did not induce disease symptoms on their host of isolation. However, the strains induced hypersensitive reaction on tobacco, geranium, melon, squash, and tomato leaves via leaf infiltration. Multilocus sequence analysis suggested that the strains belong to clade IA of *Xanthomonas*, phylogenetically close to *Xanthomonas translucens*, *X. theicola*, and *X. hyacinthi*. Average nucleotide identity and digital DNA-DNA hybridization values between the whole-genome sequences of the strains isolated in this study and reference *Xanthomonas* strains are far below the accepted thresholds for the definition of prokaryotic species, signifying that these strains could be defined as two new species within clade IA of *Xanthomonas*. Comparative genomics showed that the strains isolated from crown gall tissues are genetically distinct from *X. translucens*, as almost all the type III secretion system genes and type III effectors are lacking in the former group. The data obtained in this study provide novel insight into the breadth of genetic diversity of crown gall-associated bacteria and pave the way for research on gall-associated *Xanthomonas*-plant interactions.

**IMPORTANCE** Tumorigenic agrobacteria—members of the bacterial family *Rhizobiaceae*—cause crown gall and hairy root diseases on a broad range of plant species. These bacteria are responsible for economic losses in nurseries of important fruit trees and ornamental plants. The microclimate of crown gall and their accompanying microorganisms has rarely been studied for the microbial diversity and population dynamics of gall-associated bacteria. Here, we employed a series of biochemical tests, pathogenicity assays, and molecular-phylogenetic analyses, supplemented with comparative genomics, to elucidate the biological features, taxonomic position, and genomic repertoires of five crown gall-associated *Xanthomonas* strains isolated from weeping fig and *Amaranthus* sp. plants in Iran. The strains investigated in this study induced hypersensitive reactions (HR) on geranium, melon, squash, tobacco, and tomato leaves, while they were nonpathogenic on their host of isolation. Phylogenetic analyses and whole-genome-sequence-based average

**Ad Hoc Peer Reviewer** Michael O'Leary

Address correspondence to Nemanja Kuzmanović, nemanja.kuzmanovic@julius-kuehn.de, or Ebrahim Osdaghi, eosdaghi@ut.ac.ir.

*Present address: Nemanja Kuzmanović, Julius Kühn-Institut, Federal Research Centre for Cultivated Plants, Institute for Plant Protection in Horticulture and Forests, Braunschweig, Germany.

The authors declare no conflict of interest.

nucleotide identity (ANI)/digital DNA-DNA hybridization (dDDH) calculations suggested that the *Xanthomonas* strains isolated from crown gall tissues belong to two taxonomically unique clades closely related to the clade IA species of the genus, i.e., *X. translucens*, *X. hyacinthi*, and *X. theicola*.

**KEYWORDS** *Agrobacterium* sp., *Amaranthus* sp., clade I *Xanthomonas*, *Ficus benjamina*, T3SS, weeping fig

Crown gall, a bacterial plant disease caused by tumorigenic agrobacteria (i.e., *Agrobacterium* spp., *Allorhizobium* spp., and *Rhizobium* spp.), is one of the economically most important constraints in commercial nurseries and the ornamental-plant industry (1). Crown gall symptoms include overgrowth and hyperplasia in the young tissues of plants due to infection with agrobacterial strains harboring the tumor-inducing (Ti) plasmid. Transfer of the T-DNA fragment of the Ti plasmid into the plant cells transforms them to synthesize plant hormones, i.e., indole-3-acetic acid (IAA; auxin) and cytokinin, and initiates gall formation. The imbalanced status of phytohormones inside the plant tissues leads to overgrowth and gall formation at the site of infection (2). Genetically transformed plant cells within the gall tissue are also responsible for synthesizing strain-specific nutrient sources called opines to be used by the gall-inducing pathogen. The nutrient-rich environment of crown gall tissue is a sink of amino acids and sugars or organic acids that can also be used as nutrients for the growth of several bacterial groups other than the pathogenic agrobacteria. Hence, gall tissue is a suitable environment to be colonized by epiphytic and endophytic populations of bacterial species (3).

During the past decade, severe outbreaks of crown and stem gall diseases were observed in commercial nurseries and ornamental-plant greenhouses in southern Iran. Several agrobacterial strains were isolated from symptomatic plants and investigated using molecular-phylogenetic approaches. As a result, dozens of *Agrobacterium* spp. and *Allorhizobium* species strains which were isolated from gall tissues were shown to cause the same symptoms on the host plants under greenhouse conditions (4). However, we have simultaneously isolated a number of atypical bacterial strains from the gall issues of weeping fig (*Ficus benjamina*) and *Amaranthus* sp. plants in the Fars province of southern Iran. Preliminary examinations, i.e., Gram reaction, colony color, and morphology on culture media, revealed that the gall-associated atypical bacterial strains resemble members of the genus *Xanthomonas* (5).

*Xanthomonas* constitutes a group of plant-associated or plant-pathogenic bacterial strains capable of infecting taxonomically diverse plant species. These bacteria cause devastating diseases on hundreds of annual crops, fruit trees and ornamental species, including economically important crops within the families *Cucurbitaceae*, *Fabaceae*, *Poaceae*, *Rosaceae*, and *Solanaceae* (6, 7). Currently, the genus comprises 31 validly described species, some of which have several subspecies and pathovars (8). Plant-pathogenic xanthomonads use a type III secretion system (T3SS) to translocate a cocktail of different effector proteins into host plant cells, referred to as type III effectors (T3Es). These T3Es are generally classified into transcription activator-like effectors (TALEs) and non-TALEs, with the latter group also being known as *Xanthomonas* outer proteins (Xops). Nucleic acid sequence-based phylogenetic analyses suggest that xanthomonads can be divided into two distinct clades; clade I includes the economically important small-grain-cereal pathogen *Xanthomonas translucens*, as well as a number of less important species, such as *X. albilineans*, *X. hyacinthi*, and *X. theicola* (9), whereas clade II xanthomonads include most of the well-studied plant pathogens, i.e., *X. campestris*, *X. oryzae*, and *X. arboricola*, along with the former *X. axonopodis* species complex, which includes *X. citri* and *X. euvesicatoria* (6, 10).

The purposes of the present study were to identify and characterize the *Xanthomonas*-like bacterial strains isolated from crown gall tissues of weeping fig and *Amaranthus* sp. plants in Iran. With this aim, we conducted a series of inoculation

assays on the host of isolation as well as a number of taxonomically diverse plant species under greenhouse conditions. Multilocus sequence analysis (MLSA) of four house-keeping genes was also conducted to determine the phylogenetic position of the strains within the *Xanthomonas* species. Furthermore, whole-genome-sequence-based comparisons and comparative genomics were used to elucidate the exact taxonomic position, genomic features, and virulence-associated repertories of the strains. Taken together, phenotypic and molecular-phylogenetic analyses revealed that the strains associated with weeping fig and *Amaranthus* sp. crown gall tissues belong to two taxonomically unique clades closely related to the clade IA species of the genus, i.e., *X. translucens*, *X. hyacinthi*, and *X. theicola*.

## RESULTS

**Microbiological features of the bacterial strains.** Five bacterial strains (Table 1) which were morphologically distinct from the crown gall pathogenic agrobacteria were isolated from crown gall tissues of weeping fig and *Amaranthus* sp. plants in Shiraz (Iran) in 2019. Pale yellow, domed, circular colonies 1 to 2 mm in diameter were observed on general culture media, i.e., nutrient agar (NA) and yeast-extract peptone glucose agar (YPGA), after 48 to 72 h of incubation. All the strains were Gram negative, oxidase negative, and catalase positive and had oxidative metabolism. All the strains had amylolytic and pectolytic activity and were able to hydrolyze esculin, Tween 20/80, gelatin, and casein. They were able to utilize cellobiose, maltose, inositol, galactose, fructose, arabinose, xylose, and sorbitol. They were negative for utilization of raffinose, while all the reference strains of xanthomonads were positive for this feature. The strains produced pale yellow and mucoid colonies on yeast extract-dextrose-calcium carbonate (YDC) agar medium, characteristic of *Xanthomonas* spp. Based on the morphological features and results of biochemical tests, the bacterial strains were suspected to belong to the genus *Xanthomonas*.

**Pathogenicity and plant colonization of the strains.** The bacterial strains were evaluated for pathogenicity on their host of isolation as well as a set of taxonomically diverse annual crops, vegetables, and weeds using three inoculation techniques, i.e., spraying, leaf infiltration, and needle prick. All the plant species inoculated with spraying and needle prick techniques remained asymptomatic until 20 days postinoculation (dpi). Thus, the gall-associated *Xanthomonas*-like bacteria did not induce symptoms on the plants when they were inoculated alone. However, when the strains isolated in this study were coinoculated on sunflower plantlets along with the *Agrobacterium larrymoorei* strain Ficamol, the former group did not affect the pathogenicity of crown gall pathogen, and *Agrobacterium*-induced crown gall symptoms were observed on the inoculated plants 20 to 30 dpi (Fig. 1A to C). In the leaf infiltration method, a hypersensitivity reaction (HR), i.e., necrotic areas with a blackish brown appearance, was observed at the site of inoculation on squash (Fig. 1D), melon (Fig. 1E), and tomato (Fig. 1F) as well as tobacco and geranium leaves 36 to 48 hpi. The same HR symptoms were observed on tomato and squash leaves infiltrated with *X. translucens* pv. *translucens* ICMP 5752$^T$ (Fig. 1G and H; Table 1). Reisolation of the bacterial strains from crown gall tissues of the plants which were coinoculated with *A. larrymoorei* and *Xanthomonas* sp. revealed a lower proportion of *Xanthomonas* colonies than *Agrobacterium* colonies on culture media (data not shown).

Furthermore, in order to provide precise insight into the biology of the strains isolated in this study, *in planta* growth of strains AmX2 and FX1 was investigated on 12 plant species, i.e., *Amaranthus* sp., barley, black nightshade (*Solanum nigrum*), red nightshade (*Solanum villosum*), common bean, common zinnia (*Zinnia elegans*), maize, melon, spinach (*Spinacia oleracea* L. cv. Hudson), squash, watermelon, and wheat, under greenhouse conditions. Population sizes of the inoculated bacterial strains were determined 7, 14, 21, and 28 dpi and presented in log CFU/g of the leaf tissue (Fig. 1I and J). On the seventh day of inoculation, populations of both FX1 and AmX2 strains were smaller on spinach, black nightshade, and red nightshade than on the other plant species, while the largest populations of FX1 and AmX2 were observed on wheat and barley leaves, respectively. The same patterns were observed 14, 21, and 28 dpi; populations of both strains subsided over this period of time on spinach, black nightshade,

**TABLE 1** Biochemical characteristics and pathogenicity of the *Xanthomonas* strains investigated in this study[a]

| Strain | Pectolytic activity | Amylolytic activity | Utilization of: | | Virulence on: | | | | | | | | | | | | | |
| --- | --- | --- | --- | --- | --- | --- | --- | --- | --- | --- | --- | --- | --- | --- | --- | --- |
| | | | Raffinose | Mannitol | Barley | Common bean | Cucumber | Maize | Melon | Squash | Sunflower | Tomato | Watermelon | Wheat | Tobacco | Geranium |
| *Xanthomonas* sp. strain FX1 | +[b] | + | − | + | − | − | − | − | * | * | − | * | − | − | * | * |
| *Xanthomonas* sp. strain FX2 | + | + | − | + | − | − | − | − | * | * | − | * | − | − | * | * |
| *Xanthomonas* sp. strain FX3 | + | + | − | + | − | − | − | − | * | * | − | * | − | − | * | * |
| *Xanthomonas* sp. strain FX4 | + | + | − | + | − | − | − | − | * | * | − | * | − | − | * | * |
| *Xanthomonas* sp. strain AmX2 | + | + | + | + | − | − | − | − | * | * | − | * | − | − | * | * |
| *X. translucens* pv. translucens ICMP 5752^T | − | + | + | + | + | ND | ND | ND | ND | * | ND | * | ND | − | ND | ND |
| *X. translucens* pv. undulosa ICMP 11055 | − | + | + | + | + | ND | ND | ND | ND | ND | ND | ND | ND | + | + | + |
| *X. arboricola* pv. pruni ICMP 17186 | − | + | + | − | ND | ND | ND | ND | ND | ND | ND | + | ND | ND | + | + |
| *X. arboricola* pv. juglandis CFPB 8608 | − | + | + | − | ND | ND | ND | ND | ND | ND | ND | + | ND | ND | + | + |
| *X. citri* subsp. *citri* SU261 | − | + | + | + | ND | ND | − | ND | ND | ND | ND | ND | ND | ND | ND | ND |
| *X. euvesicatoria* pv. *euvesicatoria* ICMP 22075 | − | − | + | + | ND | ND | − | ND | ND | ND | ND | + | ND | ND | ND | ND |
| *X. gardneri* ICMP 16689 | − | − | + | + | ND | ND | − | ND | ND | ND | ND | + | ND | ND | ND | ND |
| *X. alfalfae* subsp. *alfalfa* Xa119 | − | + | + | + | ND | + | ND | ND | ND | ND | ND | − | ND | ND | + | ND |

[a]ND, not determined; *, induced an HR on these plants.

[b]+, indicates positive; −, indicates negative results.

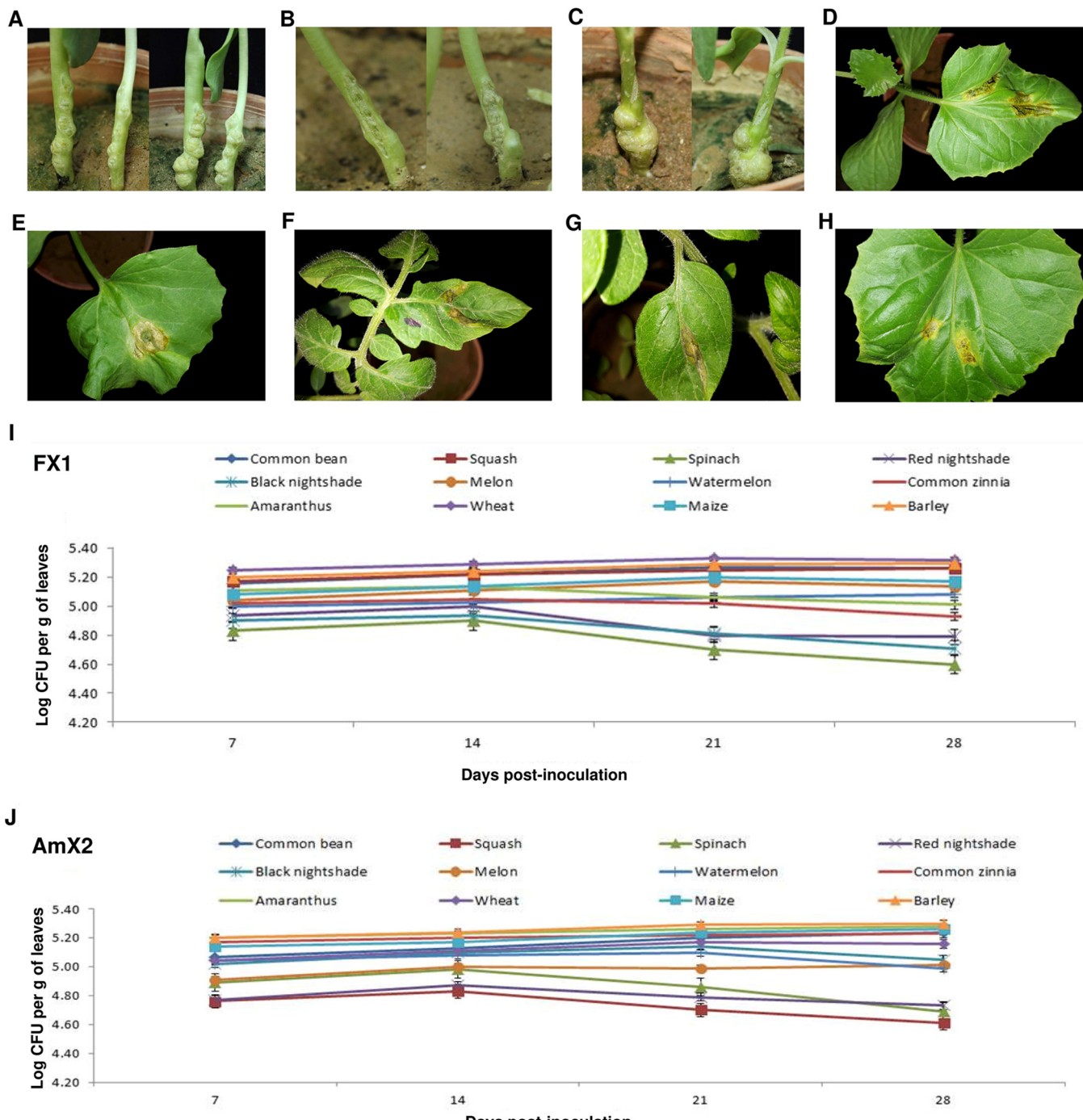

**FIG 1** Crown gall symptoms induced on sunflower plants inoculated with *Agrobacterium larrymoorei* strain Ficamol alone (A), Ficamol plus *Xanthomonas* strain FX1 (B), and Ficamol plus *Xanthomonas* strain AmX2 (C) under greenhouse conditions 14 (left) and 21 (right) days postinoculation. Leaf infiltration of squash (D), melon (E), and tomato (F) plants using bacterial suspension of the strain FX1 led to a hypersensitive reaction (HR) (necrotic areas with a blackish brown appearance) at the site of inoculation 36 to 48 hpi. The same symptoms were observed when tomato (G) and squash (H) leaves were infiltrated with *X. translucens* pv. translucens ICMP 5752$^{T}$ as control. *In planta* bacterial population sizes of *Xanthomonas* strains FX1 (I) and AmX2 (J) on 12 annual crops and weed species under greenhouse conditions, recorded and presented in log CFU/g of the leaf tissue at 7, 14, 21, and 28 days postinoculation. Wheat and spinach plants showed the highest and lowest population of the strains, respectively, in all the time frames. Error bars indicate standard deviations.

and red nightshade to the lowest level at 28 dpi. On the other hand, no significant variation was observed in the population count of the strains on barley, common bean, maize, squash, and wheat over time. The highest population over the period of 28 days was observed on wheat plants inoculated with strain FX1 (Fig. 1I and J).

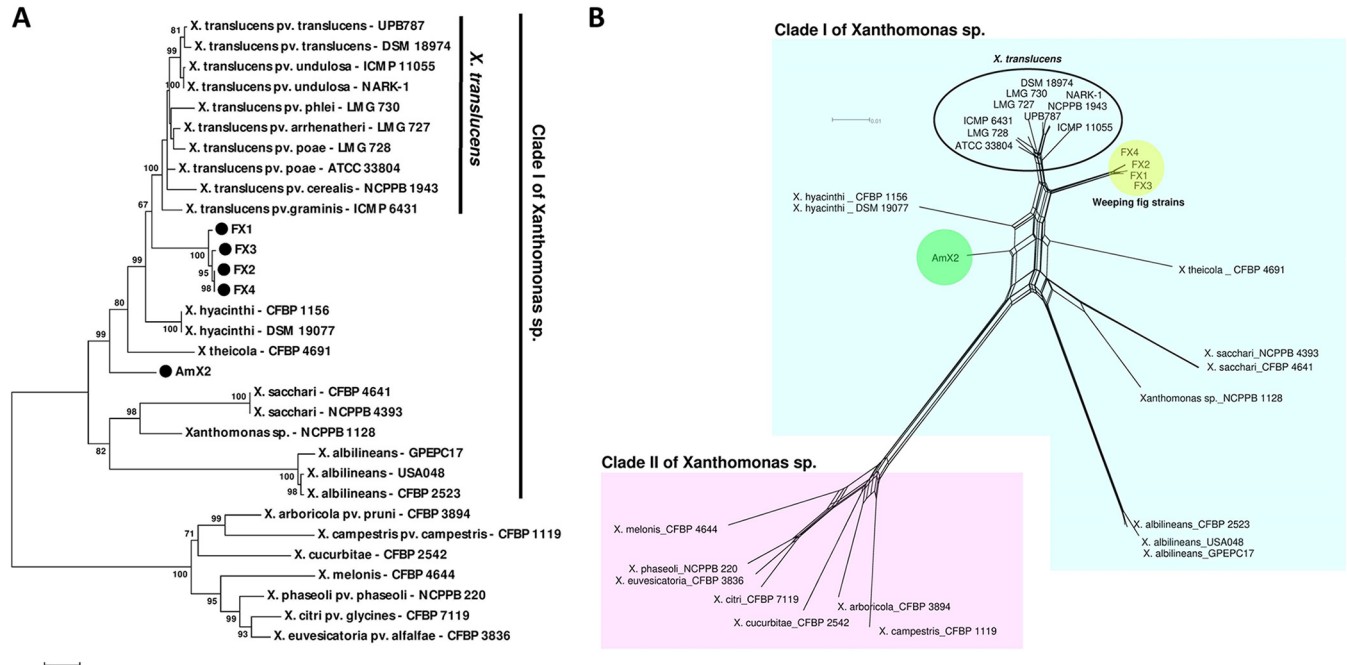

**FIG 2** Phylogeny of crown gall-associated *Xanthomonas* strains isolated in this study based on the concatenated sequences of four housekeeping genes, i.e., *atpD*, *efp*, *gyrB*, and *rpoD*, using tree-based maximum-likelihood (A) and network-based NeighborNet (B) analyses. The five *Xanthomonas* strains isolated from weeping fig and *Amaranthus* sp. were divided into two clades, where the weeping fig strains were closely related to but still distinct from the *X. translucens* species complex. The *Amaranthus* sp. strain was distinct from all the validly described species with clade I of *Xanthomonas*. Further, parallel lines/branches within the nodes of the NeighborNet network indicate possible recombination between the connected strains.

**Molecular-phylogenetic analyses.** The five *Xanthomonas*-like strains were subjected to a set of molecular-phylogenetic analyses to elucidate their precise taxonomic position. *Xanthomonas*-specific PCR primers Xc-lip-F2 and Xc-lip-R2 directed the amplification of a 777-bp DNA fragment in all the strains, confirming their identity as members of this genus. However, none of the species-specific PCR primers designed for detection of different plant-pathogenic *Xanthomonas* species/pathovars (see Table S1 in the supplemental material) could amplify the expected DNA fragment, leaving the species status of the strains undetermined. A BLAST search on the NCBI GenBank database using the individual sequences of four housekeeping genes, i.e., *atpD*, *efp*, *gyrB*, and *rpoD*, revealed that the five strains had variable sequence similarity (93 to 99% identity) to different species within clade I of *Xanthomonas*, i.e., *X. translucens*, *X. hyacinthi*, *X. theicola*, and *X. sacchari* (data not shown). MLSA-based phylogenetic investigations using the concatenated sequences of the four genes showed that the strains isolated from weeping fig were clustered in a monophyletic clade close to the small-grain-cereal pathogen *X. translucens*, while strain AmX2, isolated from *Amaranthus* sp., clustered distinctly from all the species within clade I (Fig. 2A). Because maximum-likelihood phylogeny can present conflicting topologies, a phylogenetic network was generated using the NeighborNet method (11) for the concatenated data set of sequences (Fig. 2B). The phylogenetic network confirmed divergence of the weeping fig strains from the *X. translucens* species complex, showing that the two groups were clustered in separate clades. The closest species to strain AmX2 was *X. hyacinthi*, while the two clades were too distinct to be considered members of the same species (Fig. 2B).

**Genome sequencing and overall genome relatedness.** Considering the MLSA results, three representative strains, i.e., FX1, FX4, and AmX2, were subjected to whole-genome sequencing using the Illumina HiSeq X platform and assembled using the SPAdes genome assembler (12). Quality assessment of the genome assembly and annotation using the BUSCO online service (13) confirmed the accuracy and completeness of all three genomes compared to the complete genome sequence of *X. translucens* DSM 18974. Details of the genome quality assessment criteria are shown in Table S2A. The genome sizes of the strains were 5,383 kbp (in 152 contigs [FX1]), 5,383 kbp

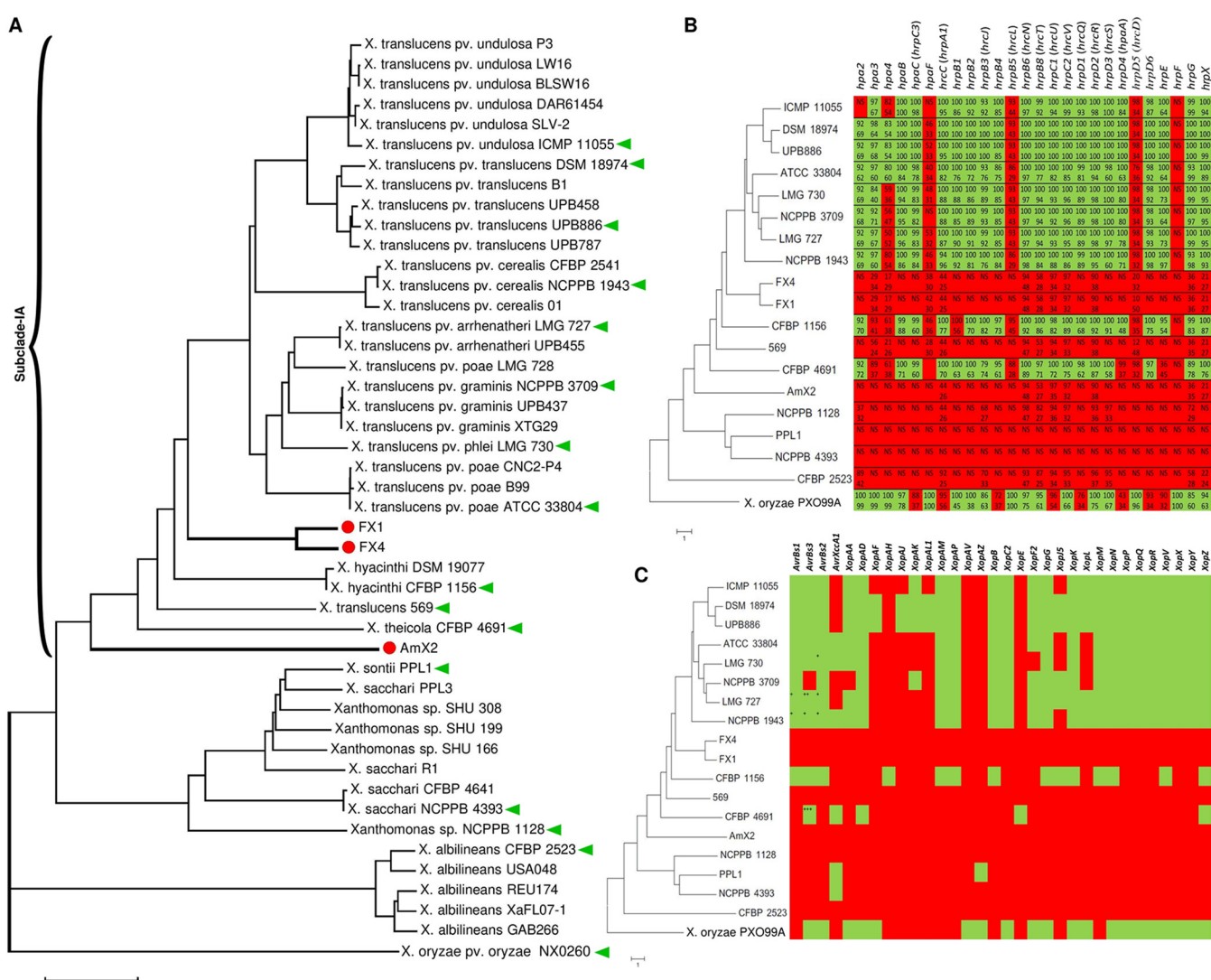

**FIG 3** Average nucleotide identity (ANI)-based neighbor-joining distance clustering tree of all available clade I *Xanthomonas* strains constructed using the ANI/AAI-Matrix genome-based distance matrix calculator (A). The clade I xanthomonads were divided into two subclades (9), where strains FX1, FX4, and AmX2, which were sequenced in this study, were clustered in subclade IA. Predicted type III secretion system (T3SS) (B) proteins and *Xanthomonas* outer proteins (Xops or non-TALEs) (C) in the three *Xanthomonas* strains sequenced in this study were compared to a representative set of clade I *Xanthomonas* genomes retrieved from the NCBI GenBank. (A) Red circles indicate the three strains sequenced in this study; green triangles indicate the strains included in the comparative genomics analyses. (B and C) Green and red squares indicate the presence and absence of the respective genes, respectively, while the numbers inside the squares (B) indicate the query coverage (top numbers) and sequence similarity (bottom numbers) indices in the BLASTp search. NS, no significant similarity was found in BLASTp.

(in 89 contigs [FX4]), and 5,069 kbp (in 75 contigs [AmX2]), with the GC content ranging from 68.8% to 69.8%, which is within the typical range of clade I *Xanthomonas* strains. Based on the annotation obtained with the NCBI Prokaryotic Genome Annotation Pipeline (PGAP) (14), strains FX1, FX4, and AmX2 contain a total of 4,502, 4,468, and 4,170 coding genes, respectively, with 40, 68, and 41 pseudogenes, respectively. Genomic features of the strains sequenced in this study in comparison to the phylogenetically closely related species are shown in Tables S3 and S4.

A genome-based distance matrix calculator (ANI/AAI-Matrix) was used to conduct average nucleotide identity (ANI) distance clustering of genome sequences obtained in this study, first for all 32 reference strains of *Xanthomonas* spp. and then for the representative species of clade I xanthomonads available in the NCBI GenBank database. The resulted neighbor-joining tree revealed distinct clustering of strains FX1 and FX4 and the other clade IA xanthomonads. The former strains were clustered in a monophyletic clade closely related

Microbiology
Spectrum

**TABLE 2** ANI and dDDH values among the representative set of clade I *Xanthomonas*, including the strains sequenced in this study[a]

| | Taxon | Strain | 1 | 2 | 3 | 4 | 5 | 6 | 7 | 8 | 9 | 10 | 11 | 12 | 13 | 14 | 15 | 16 | 17 | 18 |
|---|---|---|---|---|---|---|---|---|---|---|---|---|---|---|---|---|---|---|---|---|
| 1 | *Xanthomonas* sp. | FX1 | | 83.8 | 33.9 | 50.2 | 48.4 | 48.7 | 48.6 | 47.9 | 49.6 | 49.6 | 50.9 | 50.3 | 47.2 | 41.3 | 32.5 | 32.9 | 27.6 | 32.7 |
| 2 | *Xanthomonas* sp. | FX4 | 98.1 | | 34.0 | 50.4 | 48.5 | 48.6 | 48.5 | 47.9 | 49.6 | 49.5 | 50.8 | 49.9 | 47.2 | 41.3 | 32.5 | 32.9 | 27.6 | 32.6 |
| 3 | *Xanthomonas* sp. | AmX2 | 87.7 | 87.7 | | 34.0 | 33.5 | 33.6 | 33.5 | 33.7 | 33.8 | 33.7 | 34.6 | 34.6 | 34.3 | 33.6 | 30.7 | 30.9 | 26.1 | 30.7 |
| 4 | *X. translucens* pv. *graminis* | NCPPB 3709 | 93.1 | 93.1 | 87.8 | | 62.8 | 63.5 | 63.4 | 59.7 | 79.9 | 74.0 | 69.2 | 48.8 | 46.4 | 40.1 | 32.6 | 32.8 | 27.3 | 32.8 |
| 5 | *X. translucens* pv. *undulosa* | ICMP 11055 | 92.7 | 92.7 | 87.6 | 95.5 | | 78.5 | 77.8 | 60.3 | 62.3 | 62.6 | 62.3 | 47.0 | 44.8 | 39.2 | 32.0 | 32.4 | 26.7 | 32.3 |
| 6 | *X. translucens* pv. *translucens* | UPB886 | 92.7 | 92.7 | 87.6 | 95.6 | 97.7 | | 89.9 | 59.7 | 62.8 | 63.3 | 62.7 | 47.0 | 45.0 | 39.5 | 32.2 | 32.6 | 26.8 | 32.4 |
| 7 | *X. translucens* pv. *translucens* | DSM 18974 | 92.8 | 93.0 | 87.7 | 95.7 | 97.6 | 98.9 | | 59.7 | 62.5 | 62.9 | 62.5 | 47.0 | 44.9 | 39.4 | 32.2 | 32.5 | 26.7 | 32.4 |
| 8 | *X. translucens* pv. *cerealis* | NCPPB 1943 | 92.7 | 92.7 | 87.6 | 94.9 | 94.8 | 94.4 | 94.9 | | 59.8 | 59.4 | 59.2 | 47.1 | 44.8 | 39.4 | 32.2 | 32.3 | 27.3 | 32.5 |
| 9 | *X. translucens* pv. *arrhenatheri* | LMG 727 | 92.9 | 92.9 | 87.8 | 97.7 | 95.6 | 95.6 | 95.7 | 94.8 | | 76.3 | 69.1 | 48.2 | 46.0 | 40.2 | 32.3 | 32.5 | 26.8 | 32.4 |
| 10 | *X. translucens* pv. *phlei* | LMG 730 | 93.0 | 93.0 | 87.8 | 97.3 | 95.2 | 95.8 | 95.5 | 94.8 | 97.2 | | 69.1 | 47.6 | 45.9 | 39.9 | 32.3 | 32.5 | 27.0 | 32.4 |
| 11 | *X. translucens* pv. *poae* | ATCC 33804 | 93.4 | 93.4 | 87.8 | 96.5 | 95.1 | 95.5 | 95.4 | 94.7 | 96.5 | 96.1 | | 48.7 | 46.8 | 41.1 | 33.0 | 33.3 | 28.0 | 33.3 |
| 12 | *X. hyacinthi* | CFBP 1156 | 92.9 | 92.8 | 87.9 | 92.5 | 92.9 | 92.3 | 92.5 | 92.1 | 92.5 | 92.4 | 92.5 | | 49.5 | 41.9 | 33.3 | 33.8 | 27.6 | 33.6 |
| 13 | *X. translucens* | 569 | 92.5 | 92.6 | 88.2 | 92.4 | 91.7 | 91.8 | 91.9 | 91.7 | 92.0 | 92.0 | 92.4 | 92.8 | | 43.5 | 33.2 | 33.7 | 27.8 | 33.5 |
| 14 | *X. theicola* | CFBP 4691 | 90.6 | 90.6 | 87.6 | 90.2 | 89.6 | 89.7 | 89.8 | 89.7 | 90.2 | 89.9 | 90.3 | 90.5 | 91.4 | | 32.2 | 32.6 | 27.1 | 32.4 |
| 15 | *Xanthomonas* sp. | NCPPB 1128 | 87.0 | 87.0 | 86.1 | 86.8 | 86.7 | 86.7 | 87.0 | 86.6 | 86.7 | 86.7 | 87.3 | 87.1 | 87.5 | 86.6 | | 50.5 | 28.2 | 52.1 |
| 16 | *X. sacchari* | NCPPB 4393 | 87.3 | 87.3 | 86.1 | 87.2 | 86.8 | 86.7 | 87.2 | 86.7 | 87.1 | 86.8 | 87.3 | 87.5 | 87.6 | 87.0 | 92.9 | | 28.8 | 55.4 |
| 17 | *X. albilineans* | CFBP 2523 | 83.6 | 83.5 | 82.5 | 93.5 | 83.3 | 83.2 | 83.5 | 83.4 | 83.2 | 83.4 | 83.6 | 83.6 | 83.8 | 83.5 | 84.2 | 84.2 | | 28.5 |
| 18 | *X. sontii* | PPL1 | 87.3 | 87.0 | 86.1 | 86.8 | 86.7 | 86.7 | 86.9 | 86.6 | 86.8 | 86.8 | 87.3 | 87.4 | 87.7 | 86.6 | 93.6 | 94.0 | 84.2 | |

[a]A combination of ANI and dDDH indices was used to designate a taxonomic status for a given phylogenetic clade; strains FX1 and FX4 were suspected to belong to a novel species, while strain AmX2 could be considered another novel species. Highlighted squares indicate ANI (below the diagonal) and dDDH (above the diagonal) values higher than the accepted threshold for the definition of prokaryotic species.

to but still distinct from the *X. translucens* species complex. The strain AmX2 was also distinct from the closest neighbor species, i.e., *X. hyacinthi* (Fig. 3A). In order to estimate DNA similarity indexes between the strains sequenced in this study and all *Xanthomonas* species, ANI values were calculated using three online services—JSpeciesWS, ANI Calculator, and OrthoANIu (15–17) (http://enve-omics.ce.gatech.edu/g-matrix/)—and the ultimate ANI value was obtained via combination and averaging of the results generated by the three online services.

According to the MLSA results, which confirmed taxonomic position of the strains isolated in this study in clade I of *Xanthomonas* spp., type strains of all clade I *Xanthomonas* species were included in DNA similarity analyses. The resulting data showed that the strains FX1 and FX4 had only 83.6% to 93.1% sequence identity with the type strains of previously described *Xanthomonas* species, while the ANI value between strain AmX2 and the other *Xanthomonas* species was less than 90.0% (Table 2). On the other hand, the digital DNA-DNA hybridization (dDDH) value (18) among the strains sequenced in this study and those of the reference strains was consistently lower than 60.0%. These ANI and dDDH values are far below the accepted threshold for prokaryotic species definition, i.e., ≤95% and ≤70% for ANI and dDDH, respectively, suggesting that strains FX1, FX4, and AmX2 belong to two new species within clade IA of *Xanthomonas*. While strains FX1 and FX4 belong to the same species, strain AmX2 could be considered another standalone species distinct from all validly described *Xanthomonas* spp. (Fig. 3A). A formal comprehensive polyphasic taxonomic study is warranted to further refine the position of these strains within the genus.

**Comparative genomics.** It has been shown that clade I *Xanthomonas* species possess high genetic diversity in terms of the T3SS, T3Es, and TALEs. Genome-informed investigation of *Xanthomonas* spp. would provide valuable information on the virulence repertories, pathogenicity mechanisms, and host adaptation of the strains (9). Furthermore, genomic investigation of the *Xanthomonas* strains associated with crown gall tissues would decipher their potential unique capabilities that support their survival in the gall environment. Significant variations were observed among the clade 1 xanthomonads in their T3SS repertories (Fig. 3B). When a 60% cutoff index was used for both query coverage and sequence similarity criteria, no T3SS gene was detected in the three strains sequenced in this study in BLASTp-based investigations (http://www.ncbi.nlm.nih.gov/blast/). However, a number of suspected sequences with high query coverage but low sequence similarity were

predicted in strains AmX2, FX1, and FX4 (Fig. 3B). For instance, a sequence similar to *hrpB6* (94% query coverage and 48% sequence similarity) was predicted in AmX2, FX1, and FX4. *hrpB6* is a T3SS ATPase in *X. euvesicatoria* (gene ID: 61777304). Furthermore, *hrpC2* (originally described in *X. euvesicatoria*; accession number P80150), which belongs to the FHIPEP (flagella/HR/invasion protein export pore) family, was detected in all the three strains with 97% query coverage and 32% similarity. The T3SS pattern in the strains AmX2, FX1, and FX4 was similar to those of *X. translucens* strain 569, *Xanthomonas* sp. strain NCPPB 1128, "*X. sontii*" strain PPL1, *X. sacchari* strain NCPPB 4393, and *X. albilineans* strain CFBP 2523, while members of the *X. translucens* species complex carried most of the T3SS genes (Fig. 3B).

As for the *Xanthomonas* outer proteins (Xops or non-TALEs), *in silico* analyses revealed that the three strains sequenced in this study encode none of the Xops included in this investigation. The same pattern was observed in *X. translucens* 569, *Xanthomonas* sp. strain NCPPB 1128, and *X. albilineans* CFBP 2523, which was significantly different from that observed in the *X. translucens* species complex (Fig. 3C). The protein XopAV was absent in all the clade I xanthomonads investigated in this study, while XopAH, XopE, and XopAZ were present only in *X. hyacinthi* CFBP 1156, *X. theicola* CFBP 4691, and "*X. sontii*" strain PPL1, respectively. In order to visualize the T3SS and Xop repertories of the strains sequenced in this study, BRIG (19) was implemented in whole-genome-based comparisons of strains FX1, FX4, and AmX2 using *X. theicola* strain CFBP 4691 and *X. translucens* pv. *translucens* strain DSM 18974 as reference genomes (Fig. S1). Considering the satisfactory completeness of the genome sequencing and assembly of the three gall-associated strains (Table S2A), circular mapping against the reference genomes CFBP 4691 (Fig. S1A) and DSM 18974 (Fig. S1B) confirmed the absence of the T3SS and *xop* genes in the three sequences obtained in this study.

Comparative genomics using the data provided by the RAST (20) and SEED-Viewer (21) databases revealed several lines of evidences highlighting differences in metabolic networks and subsystem profiles between the three strains sequenced in this study and those of the clade I xanthomonads, as well as the reference strain X. *oryzae* pv. oryzae PXO99[A] (Fig. 4A). For instance, the number of subsystems that are responsible for biotin biosynthesis, invasion and intracellular resistance, siderophore production, protein biosynthesis, nitrogen metabolism, central aromatic intermediate metabolism, sulfur metabolism, methylglyoxal metabolism, potassium metabolism, maltose and maltodextrin utilization, respiration, and the miscellaneous subsystems in strains FX1 and FX4 was higher than in the members of their neighbor clade *X. translucens*. The number of subsystems responsible for tetrapyrroles, capsular and extracellular polysaccharides, ABC transporters, protein biosynthesis, and sulfur metabolism in the strain AmX2 was higher than in the other genomes investigated. In the genome of *X. oryzae* pv. oryzae, which belongs to clade II of the xanthomonads, numbers of subsystems related to riboflavin biosynthesis, T3SS, amino sugars, chitin and *N*-acetylglucosamine utilization, programmed cell death, and toxin-antitoxin systems were significantly higher than in the clade I xanthomonads.

Orthologous gene clusters were determined using the OrthoVenn online service (22) through four-versus-four designations of the strains, where one of the clade I strains DSM 18974 and CFBP 4691 was used as the reference genome in each analysis (Fig. 4B and C). When *X. translucens* strain DSM 18974 was the reference, the three strains sequenced in this study shared 611 proteins besides the 2,609 proteins which were shared among all four strains. Strains AmX2, FX1, and FX4 possessed 35, 4, and 3 unique proteins, respectively, when DSM 18974 was designated the reference genome. Furthermore, FX1 and FX4 shared 584 proteins with each other while these strains shared 29 and 35 proteins, respectively, with AmX2, indicating closer genomic repertories of FX1 and FX4. On the other hand, when *X. theicola* CFBP 4691 was the reference genome, the three strains sequenced in this study showed 811 shared proteins along with 2,416 proteins which were shared within all four genomes. Strains AmX2, FX1, and FX4 possessed 37, 3, and 3 unique proteins, respectively, when CFBP 4691 was considered the reference genome (Fig. 4B and C).

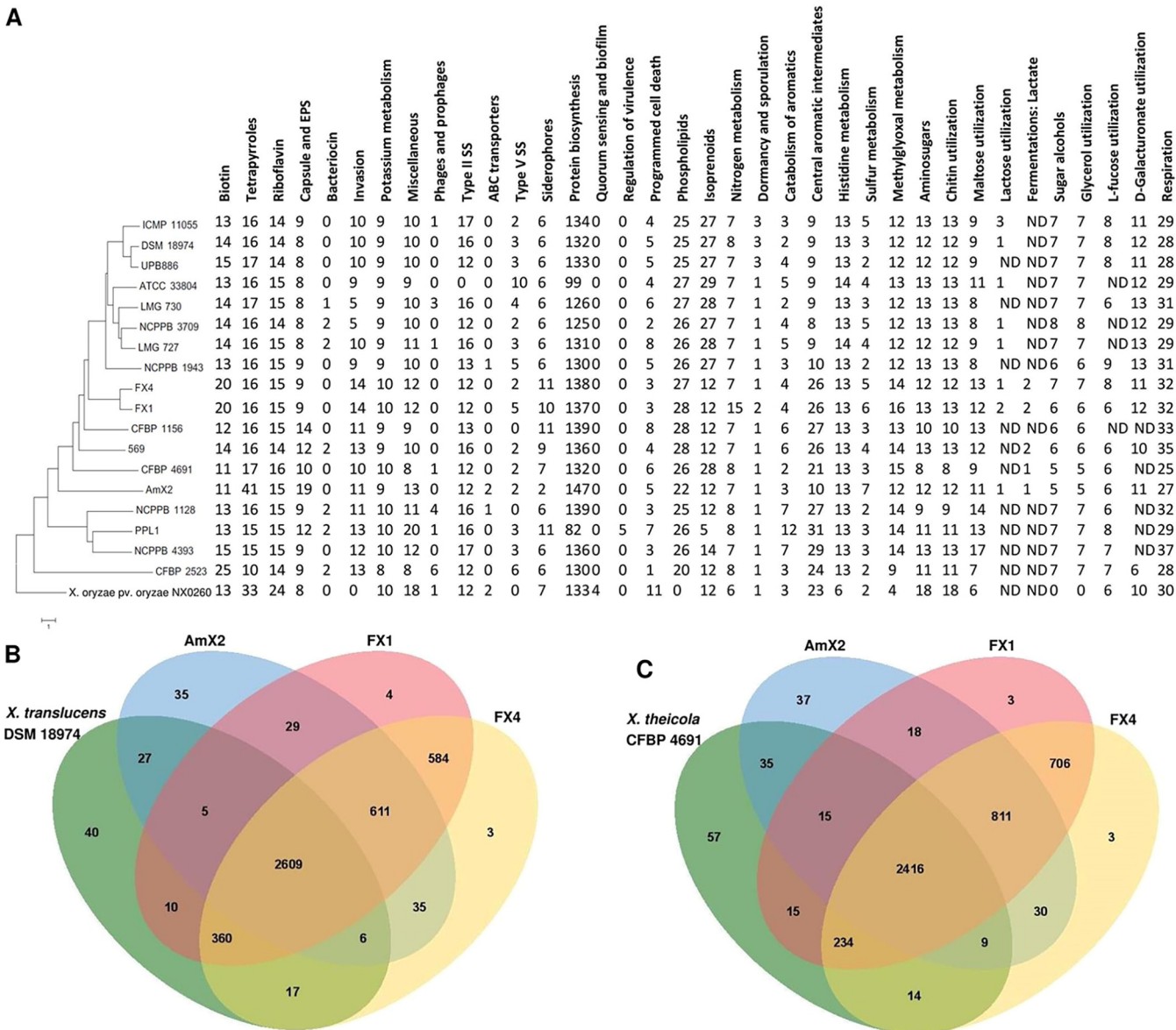

**FIG 4** Genomic characteristics of three *Xanthomonas* strains sequenced in this study compared to a set of representative clade I xanthomonads calculated using the online annotating service RAST (A). Venn diagram constructed using the OrthoVenn online service showing the shared gene families (orthologous clusters) among the three strains sequenced in this study and the two reference genomes from *X. translucens* pv. translucens DSM 18974 (B) and *X. theicola* CFBP 4691 (C).

The representative genomes of clade I xanthomonads along with the three genomes obtained in this study were screened using the web-based tool BAGEL4 (23) for the presence of hypothetical bacteriocin-encoding genes/clusters and antibiotic peptides. Zoocin A was detected in strains FX1, FX4, and AmX2 as well as all the other clade I xanthomonads except *X. theicola* CFBP 4691. The sactipeptide class of microbial products was detected in FX1 but not in FX4 and AmX2. Interestingly, rhodanodin, which is a lasso peptide produced by *Rhodanobacter thiooxydans*, was detected in AmX2 but not in any other *Xanthomonas* genome evaluated in this study (Table S5). The PHASTER online service (24) was used to detect prophage sequences within the bacterial genomes. *Escherichia* phage 500465-1 was predicted in the genome of FX1, FX4, and AmX2 as well as a number of reference strains. The *Escherichia* phage 500465-1 was the only prophage predicted in FX1 and FX4, but strain AmX2 had the *Ralstonia* phage phiRSA1 in its genome along with *X. translucens* pv. undulosa strain ICMP 11055. Table S6 shows the list of prophage sequences

in the clade I xanthomonads that were missing in the strains sequenced in this study. However, it should be noted that due to the draft nature of the genomes, it is possible that more phage sequences are present in these strains but were not identified in the present study due to unresolved repetitive sequence.

No integrative plasmid (episome) was detected using the PlasmidFinder online service (25) in the genomes of strains in this study. Visualization of genomic islands in the whole-genome sequences of the three *Xanthomonas* sp. strains obtained in this study against the genomes of two reference strains, *X. theicola* CFBP 4691 and *X. translucens* pv. translucens DSM 18974, is shown in Fig. S2. The distribution pattern of the main genomic islands in all the three strains sequenced in this study remained constant regardless of the reference genome used for prediction. When strain CFBP 4691 was considered the reference genome, two dominant genomic islands were predicted in FX1, located between 0.9Mbp and 1.0Mbp and between 2.5Mbp and 2.6Mbp. The two islands were predicted when DSM 18974 was used as the reference, although the positions of the islands were slightly different (1.5Mbp to 1.6Mbp and 2.8Mbp to 2.9Mbp). Similarly, a genomic island was constantly predicted in FX4 when both the reference genomes were used, while the position of the island changed from 2.1Mbp to 2.2Mbp and 2.9Mbp to 3.0Mbp when CFBP 4691 and DSM 18974 were used as reference genomes, respectively (Fig. S2). The same pattern was observed in strain AmX2, where a dominant genomic island was predicted at 1.6Mbp to 1.7bp and 2.3Mbp to 2.4Mbp when CFBP 4691 and DSM 18974 were used as reference genomes, respectively. These results indicate different patterns of genomic island distribution among the strains sequenced in this study despite the fact that they belong to the same species, i.e., FX1 and FX4 (Fig. S2).

## DISCUSSION

Five *Xanthomonas* sp. strains have accidentally been isolated from crown gall tissues along with several *Agrobacterium* strains, while the latter organisms were the main causal agent of the disease. In this study, we employed a series of biochemical tests, pathogenicity assays, and MLSA-based phylogenetic analyses on five crown gall-associated *Xanthomonas* strains isolated from weeping fig and *Amaranthus* sp. plants in Iran. Subsequently, three representative strains, i.e., FX1, FX4, and Amx2, were subjected to genome sequencing and comparative genomics to elucidate their biological features, taxonomic positions, and genomic repertories. The strains investigated in this study induced a hypersensitive reaction (HR) on geranium, melon, squash, tobacco, and tomato leaves, while they were nonpathogenic on their host of isolation. MLSA-based phylogenetic analyses accomplished with whole-genome sequence-based ANI/dDDH calculations suggested that the *Xanthomonas* strains isolated from crown gall tissues could be considered two novel species within clade I of *Xanthomonas*.

No *Xanthomonas* pathogen has so far been described as colonizing *Amaranthus* sp., while two distinct xanthomonads infecting *Ficus* spp. plants are described in the literature (26, 27): angular leaf spot of ornamental *Ficus* spp. species caused by *X. campestris* pv. fici (28, 29) and angular leaf spot of pipal (*Ficus religiosa*) caused by a taxonomically undetermined *Xanthomonas* species. In fact, the latter pathogen (pipal pathogen) has originally been designated a standalone species and described as *Xanthomonas fici* in India (27). However, Young et al. (30) noted that the name "*X. fici* Jindal & Patel 1972" is illegitimate because it is antedated by "*X. fici* Cavara 1905." On the other hand, the causal agent of the bacterial blight of *Ficus elastica* in Florida (USA) was identified as *X. axonopodis* based on the nucleotide sequence of its 16S rRNA gene (31). The strains isolated in this study had no phylogenetic similarity to those of the previously described species infecting *Ficus* sp. plants. Our strains were clustered within the clade I of *Xanthomonas*, while the above-mentioned species—originally described by Young et al. (32) and Parkinson et al. (33)—are members of clade II of the genus.

A physiologically active crown gall produces specific nutrients (opines) utilizable by microorganisms, providing a rich niche for plant-associated bacteria to colonize, grow,

and form complex ecological interactions (34–37). The microbial communities involved in opine ecology are not limited to *Agrobacterium* members. Complex bacterial communities of many species coexist in crown gall tissues (38). During the past few decades, various bacterial species, i.e., *Pseudomonas* spp. and coryneform bacteria, were isolated from crown gall tissues and possess the capability of competing for opines as growth substrates (34). Hence, agrobacteria, either in the soil or in association with healthy or diseased plants, are likely in environments that require mechanisms to compete with other microbes for resources (39). The bacterial type VI secretion system (T6SS) delivers effector proteins in a contact-dependent manner to antagonize other bacteria and provides a quantifiable competitive advantage over accompanying susceptible counterparts. It has been noted that only a fraction of sequenced *Agrobacterium tumefaciens* strains carry T6SS-encoding loci. Wu et al. (39) stated that in some cases, all the members of a genomospecies appear to lack T6SS-encoding loci, whereas in other genomospecies, T6SS loci are polymorphic in terms of presence and/or absence. Since we did not generate whole-genome sequences of the agrobacterial strains isolated alongside xanthomonads, the presence and putative role of T6SS in their interaction remain undetermined.

The genome of *Pseudomonas kilonensis* strain 1855-344, which can catabolize octopine, was recently shown to contain an octopine-catabolic operon named *ooxAB* (40). Successful colonization and long-term persistence of *Agrobacterium* strains on/in crown gall tissue depend partly on competition for opine nutritive resources, either between the agrobacteria or among other microorganisms, such as pseudomonads, *Sinorhizobium* spp., coryneform bacteria, or even fungal species such as *Cylindrocarpon heteronema* and *Fusarium solani* (36, 37). It was recently shown that the microbial community of natural grapevine (*Vitis vinifera*) crown galls caused by *Allorhizobium vitis* contains more than 150 bacterial species, among which members of *Pseudomonas*, *Sphingomonas*, and *Erwinia* were more common (38). Furthermore, crown galls can harbor bacteria belonging to *Corynebacterium* or *Arthrobacter* (37). The abundance of *A. vitis* in grapevine crown gall is positively correlated with the abundance of *Novosphingobium* sp., *Xanthomonas* sp., and members of *Microbacteriaceae*, suggesting the presence of a microbial hub in the crown gall environment (3). The co-occurrence of *Xanthomonas* with *A. vitis* on crown gall tissues can be explained by the presence of high concentrations of lignin, cellulose, N-glycosylated proteins, and other cell wall precursors as well as cell wall degradation products found in bark on developing gall tissue (3, 41, 42). BLAST-based *in silico* analyses using the sequences of *ooxA*, *ooxB*, and *ooxAB* against the three genome sequences obtained in this study revealed the presence of these genes in the three *Xanthomonas* strains. As detailed in Table S2B, sequences of DNA fragments with 66 to 98% query coverage and 25 to 27% sequence similarity to *ooxA*, *ooxB*, and *ooxAB* genes were detected in the genomes of all three strains. These observations suggest the potential acquisition of opine catabolism genes by gall-associated xanthomonads, which needs further in-depth investigation.

Plant inoculation assays showed that the crown gall-associated *Xanthomonas* strains were nonpathogenic on the foliage parts of their host plant and the other plant species evaluated via spray inoculation. They also did not affect the virulence and aggressiveness of their *Agrobacterium* counterparts with which they were isolated from gall tissues. However, when the strains were inoculated onto geranium, melon, squash, tobacco, and tomato via leaf infiltration, an HR was observed 36 to 48 hpi. A similar pattern was observed in Gram-positive actinobacteria, where *Curtobacterium flaccumfaciens* strains isolated from asymptomatic solanaceous vegetables, i.e., eggplant, pepper, and tomato, were pathogenic on leguminous crops, i.e., common bean, cowpea, and mung bean, but not on solanaceous plants (43, 44). A high level of *in planta* (either epiphytic or endophytic) populations of the gall-associated *Xanthomonas* strains on barley and wheat 28 dpi revealed their potential ability to persist on small grain cereals. The gall-associated strains are taxonomically closely related to the *X. translucens* species complex, which infects gramineous plants and includes economically important pathogens around the

globe. However, neither spray inoculation nor leaf infiltration of the three gall-associated *Xanthomonas* strains induced symptoms on wheat plants under greenhouse conditions (data not shown). Hence, these gall-associated strains could epiphytically persist on non-host plants, i.e., small-grain cereals and seed materials, interfering with the quarantine inspection and pathogen survey procedure.

Epiphyte growth and persistence of different xanthomonads on nonhost plants have frequently been reported (45, 46). For instance, *X. arboricola* pv. pruni and *X. citri* subsp. *citri*, pathogens of stone fruits and citrus, respectively, multiplied on red night-shade, black nightshade, bindweed, *Chenopodium*, common bean, and wheat up to 20 dpi under greenhouse conditions. Interestingly, *X. arboricola* pv. pruni successfully proliferated on both lemon and sweet lemon up to 140 dpi, attaining a population density even higher than that of *X. citri* subsp. *citri* (47). *Xanthomonas euvesicatoria* pv. euvesicatoria strains isolated from pepper were able to colonize nightshade and common bean (48, 49). All these observations highlight the role of nonhost plants in the epidemiology of bacterial plant pathogens, which needs further attention, especially when it comes to decision-making for management of the corresponding diseases under natural field conditions.

Virulence of most plant-pathogenic xanthomonads is attributed to the function of the T3SS and a set of transcription activator-like effectors (TALEs) and non-TALES or T3Es/Xops (7, 50). Comparative genomics revealed that the crown gall-associated *Xanthomonas* strains might be adapted to a nonpathogenic lifestyle, which is reflected by the lack of pathogenicity gene clusters present in the pathogenic members of clade I (Fig. 3B and C). In spite of the absence of all T3SS and T3E repertories in the three strains sequenced in this study, the induction of HR phenotype warrants further exploration. The same results were observed by Jacobs et al. (51), who reported that *X. cannabis* strains NCPPB 3753 and NCPPB 2877 isolated from cannabis (*Cannabis sativa* L.) as well as the *Xanthomonas maliensis* strain 97M isolated from rice lacked the Hrp T3SS and did not contain any of the known T3Es while stile inducing HR on nonhost plants. Further, it has been shown that nonpathogenic *Pseudomonas* strains lacking a T3SS are common leaf colonizers of healthy plants and grow as well as or better than other *Pseudomonas syringae* strains on nonhost species without causing disease (52). These nonpathogenic *Pseudomonas* strains also typically fail to induce an HR on tobacco, unlike the *Xanthomonas* strains isolated in this study.

An in-depth comparative analysis with the other clade I xanthomonads' genomes allowed us to obtain more precise insight underlying genetic diversity of these bacteria. Subsystem analyses using RAST (20) highlighted differences in the genetic contents of clade I xanthomonads, especially regarding the differences between the cereal pathogen *X. translucens* and the strains isolated in this study. For instance, the number of subsystems that are responsible for biotin biosynthesis, invasion and intracellular resistance, siderophore production, protein biosynthesis, nitrogen metabolism, central aromatic intermediates metabolism, sulfur metabolism, methylglyoxal metabolism, potassium metabolism, maltose and maltodextrin utilization, respiration, and the miscellaneous subsystems in the strains FX1 and FX4 was higher than in the members of their neighbor clade *X. translucens*. Furthermore, prophage sequences were missing in the strains sequenced in this study, but since these are draft genomes, it is possible that more phage sequences (which are often repetitive) are present in these strains but were not identified in the present study due to unresolved repetitive sequence.

The strains sequenced in this study are phylogenetically closely related to the tea (*Camellia sinensis*) pathogen *X. theicola* and small-grain-cereal pathogen *X. translucens*, all belonging to clade I of the genus, also known as early-branching xanthomonads (9, 32, 33). Providentially, high-quality complete genome sequences of representative strains from both these clade IA pathogens are available in the NCBI GenBank database, allowing us to obtain precise comparative genomics among the early-branching group of xanthomonads (33). Recently, a complete genome sequence of the type strain of *X. theicola* CFBP 4691 was published, aiming to better understand the biology and evolution of the clade 1

xanthomonads (9). Further, Shah et al. (53) conducted a comprehensive phylogenomics and comparative genomics analysis using the complete genome sequences of 14 *X. translucens* strains and several other xanthomonads as reference genomes. Draft genome sequences of the three gall-associated *Xanthomonas* strains revealed their unique phylogenetic position, suggesting that they should be described as two novel species (Fig. 3A). Furthermore, strain 569, which is labeled as *X. translucens*, should not be included within this complex species, as the ANI values between this strain and the other members of *X. translucens* were consistently below 93%, suggesting that strain 569 cannot be considered *X. translucens*. Together, our observations indicate that a formal and comprehensive taxonomic study is warranted to address the last two taxonomic issues within *X. translucens* and further refine the classification of this complex species. While the phylogenetic position of strains FX1, FX4, and AmX2 was clarified in these analyses, only further investigations using a larger collection of strains will shed light on the genetic content, biological characteristics, and taxonomic status of crown gall-associated members of *Xanthomonas*.

## MATERIALS AND METHODS

**Phenotypic characterization of the strains.** Five yellow-pigmented Gram-negative bacterial strains (FX1 to FX4 and AmX2) (Table 1) resembling the members of *Xanthomonas* were isolated from crown gall tissues along with dozens of *Agrobacterium* strains in southern Iran (4, 54, 55). The bacterial strains were resuspended in sterile distilled water (SDW) and were stored at 4°C for further use and in 15% glycerol at −70°C for long-term purposes. All purified bacterial strains were subjected to standard biochemical and physiological tests (5). Gram reaction, oxidase and catalase activity, aerobic/anaerobic growth (O/F), and colony characteristics on YDC medium were determined. Furthermore, the strains were evaluated for their capacity to degrade starch and pectate. Type/reference strains of "*Agrobacterium fabrum*" (C58), *X. euvesicatoria* pv. euvesicatoria (ICMP 109[T]), *X. euvesicatoria* pv. perforans (ICMP 16690[T]), *X. translucens* pv. undulosa (ICMP 11055), *X. translucens* pv. translucens (DSM 18974[T]), and *X. arboricola* pv. *pruni* (ICMP 17186) were used as controls in all phenotypic and biochemical tests. The biochemical tests were repeated twice.

**Plant inoculation assays.** The hypersensitive reaction (HR) was evaluated on tobacco (*Nicotiana tabacum* cv. Turkish) and geranium (*Pelargonium graveolens*) leaves using the bacterial suspension from a 48-h-old culture on NA medium at a concentration of $10^8$ CFU/ml. The five strains were evaluated for their pathogenicity on their host of isolation, i.e., weeping fig and *Amaranthus* sp., as well as a set of taxonomically diverse annual crops, vegetables, and weeds, i.e., barley (*Hordeum vulgare* cv. Behrokh, https://link.springer.com/article/10.1007/s11032-019-1026-z, https://doi.org/10.1111/ppl.13102), common bean (*Phaseolus vulgaris* cv. Sadri), cucumber (*Cucumis sativus* cv. RS2N-NADA), maize (*Zea mays* cv. KSC703), melon (*Cucumis melo* cv. Nagham), squash (*Cucurbita pepo* cv. Elena), sunflower (*Helianthus annuus* cv. Armavirski), tomato (*Solanum lycopersicum* cv. Sunseed 6189), watermelon (*Citrullus lanatus* cv. B32), and wheat (*Triticum aestivum* cv. Behrokh, https://link.springer.com/article/10.1007/s11032-019-1026-z, https://doi.org/10.1111/ppl.13102). Plant growth conditions and inoculum preparation were as described previously (56), while inoculation was performed using spray inoculation, leaf infiltration, and needle prick techniques as described by Mafakheri et al. (4). The strains were evaluated either alone or in combination with the tumorigenic *Agrobacterium* sp. strain Ficamol, with which they were simultaneously isolated from crown gall tissues. Reference strains of the above-mentioned xanthomonads were used as positive controls, while SDW was used as the negative control. Koch's postulates were accomplished by reisolating the inoculated strains on YPGA medium from the inoculated plants showing symptoms. Confirmation of the identity of the reisolated bacteria was performed using the Gram reaction and colony morphology on YDC medium as described above.

*In planta* growth of the strains AmX2 and FX1 was investigated on 12 plant species using the procedure detailed previously (47, 56). Population size of the inoculated bacterial strains was evaluated on each of the plant species at 7, 14, 21, and 28 dpi. In brief, three subsamples from the lower, middle, and upper portions of the canopy were collected from each inoculated plant species. The leaves were combined, the weight of each leaf sample was adjusted to 2 g, and the plant tissues were macerated using a sterile mortar and pestle in 40 ml of SDW. After appropriate serial dilutions ($10^{-1}$ to $10^{-4}$), 10 $\mu$l of the resulted bacterial suspension was plated on YPGA medium (in 9-cm-diameter plates). All bacterial colonies were counted 4 days postincubation using a colony counter (CX-300; Gallenkamp, England) and the bacterial population sizes were expressed as log CFU/g of fresh leaf tissues as described previously (42). All the experiments were repeated at least once.

**Molecular-phylogenetic analyses.** In order to determine the precise taxonomic and phylogenetic position of the strains, they were subjected to PCR tests using the *Xanthomonas*-specific PCR primers Xc-lip-F2 and Xc-lip-R2 as well as several species-specific primers, as detailed in Table S1. Furthermore, multilocus sequence analysis (MLSA) using the sequences of four housekeeping genes, i.e., *atpD*, *efp*, *gyrB*, and *rpoD*, was performed on all five *Xanthomonas* strains using the procedure described by Khojasteh et al. (57) (Table S1). Bacterial DNA extraction, PCR parameters, and sequencing procedure were the same as described previously (57). Subsequently, sequences of these four genes in a collection of reference *Xanthomonas* sp. strains were retrieved from the GenBank databases and included in the phylogenetic analyses. The sequences were concatenated following the alphabetic order of the genes, and a

phylogenetic tree was constructed using the maximum-likelihood method with MEGA7 software (58). The model of evolution for maximum-likelihood analysis was determined using the Modeltest tab in MEGA7, and the phylogenetic tree was constructed with bootstrapping (1,000 replications). Furthermore, SplitsTree version 4.14.4 was used to construct the NeighborNet network of the concatenated sequences of the four genes using the same set of sequences (11).

**Whole-genome sequencing and annotation.** Based on the MLSA results, the weeping fig strains FX1 and FX4 and the *Amaranthus* sp. strain AmX2 were subjected to whole-genome sequencing to further refine the taxonomic status and genomic repertoires of the strains. Total DNA of the strains was extracted using NucleoSpin microbial DNA extraction kit (Macherey-Nagel, Germany). DNA libraries were obtained with a Nextera XT DNA library prep kit (Illumina, USA). Paired-end sequencing (2 × 150 bp) was performed on an Illumina HiSeq X platform. Reads were demultiplexed using BaseSpace (Illumina). Paired-reads quality filtering and trimming were performed with the bbduk program (59). Adaptor trimming was performed using Trimmomatic (Galaxy version 0.38.1) implemented on the Galaxy web server (60). The read quality was assessed with FastQC (Galaxy version 0.72+galaxy1; http://www.bioinformatics.babraham.ac.uk/projects/fastqc/), and *de novo* sequence assembly was performed with the SPAdes genome assembler (Galaxy version 3.12.0+galaxy1) (12) using the "careful correction" option (–careful) and automatically chosen k-mer values. Short (<200-bp) and very-low-coverage (<2-fold) contigs were excluded from the final assembly. Quantitative assessment of genome assembly and annotation completeness was evaluated using BUSCO online software (13). Genome annotation was performed using the NCBI Prokaryotic Genome Annotation Pipeline with default settings (14). Additionally, the RAST server (20) was used for fully automated annotation of the bacterial genomes. Total numbers of protein-coding genes, RNA genes, and pseudogenes were determined for all the genomes.

**Whole-genome-sequence-based comparisons and comparative genomics.** MLSA-based phylogenetic analyses revealed the gall-associated strains belong to clade I of *Xanthomonas*; hence; genomic data for a representative set of 42 clade I xanthomonad strains were retrieved from the NCBI GenBank database and subjected to phylogenetic analyses. Average nucleotide identity (ANI) and digital DNA-DNA hybridization (dDDH) values were calculated among all the type/reference strains of *Xanthomonas* species using the procedure described previously (53, 61). ANI was estimated using both one-versus-one and all-versus-all strategies via different algorithms, i.e., JSpeciesWS (15), ANI Calculator (16), and OrthoANIu (17). Additionally, the Genome-to-Genome Distance Calculator (GGDC, v2.1) online service was used to calculate dDDH values among the strains (18). ANI-based all-versus-all distance matrix and the resulting neighbor-joining tree were generated using the ANI/AAI-Matrix genome-based distance matrix calculator (16) (http://enve-omics.ce.gatech.edu/g-matrix/). A combination of ANI and dDDH indices was used to assign "stand-alone species" taxonomic status to a given taxon. When both ANI and dDDH values were below the accepted threshold, i.e., ≤95% and ≤70% for ANI and dDDH, respectively, the corresponding strain was considered a potential novel species (62).

Based on the results obtained from the phylogenetic analyses, the three strains sequenced in this study were subjected to comparative genomics against several clade I *Xanthomonas* species, including the closest reference strains, *X. theicola* CFBP 4691 and *X. translucens* pv. translucens DSM 18974 (9, 63). Both tBLASTn- and BLASTp-based investigations (http://www.ncbi.nlm.nih.gov/blast/) were performed to decipher the T3SS repertories and *Xanthomonas* outer proteins (Xops) of the strains sequenced in this study in comparison to the other clade I xanthomonads. Sequences of a set of 26 T3SS and 32 Xops were retrieved from the complete genome sequences of *X. translucens* pv. translucens DSM 18974, *X. theicola* CFBP 4691, and *X. oryzae* pv. oryzae PXO99[A] and included in the BLAST searches. Amino acid sequences (BLASTp) were used in all the analyses, while nucleotide sequences (tBLASTn) were also implemented in the investigations for further confirmation. For both the BLASTp and tBLASTn, an E value of <1e−10 with 60% query coverage and 60% sequence similarity was considered the cutoff criterion, as recommended previously (64).

Genome collinearity analyses provide valuable insight into the genome arrangement and gene distribution in bacteria (65, 66). Thus, pairwise genome collinearity alignment of the three strains was performed and visualized with BRIG 0.95 using both *X. theicola* CFBP 4691 and *X. translucens* pv. translucens DSM 18974 as reference genomes (19). The annotation obtained by RAST (20) was used to reconstruct metabolic networks and subsystems. Given that the annotation procedure might affect the outcome of such comparative genomics analyses, we used the genome sequences that all annotated using the same technique as shown in Table S4. Subsequently, the genomes were transferred to the comparative environment of SEED-Viewer for comparative genomics analyses (21). The online services PlasmidFinder 2.0 (25), PHASTER (24), and BAGEL4 (23) were used to screen the genomes for the presence of integrative plasmids/episomes, prophage sequences, and bacteriocin-encoding genes/clusters, respectively. Further, genome-wide comparisons and visualization of orthologous clusters were performed using the online service OrthoVenn (22). IslandViewer 4 was used for the identification and visualization of genomic islands in the three strains sequenced in this study using *X. theicola* CFBP 4691 and *X. translucens* pv. translucens DSM 18974 as reference genomes (67).

**Data availability.** The nucleotide sequences obtained in this study were deposited into the GenBank database under the following accession numbers: *atpD*, MW222614 to MW222618; *efp*, MW222619 to MW222623; *gyrB*, MW222624 to MW222628; and *rpoD*, MW222629 to MW222633. Whole-genome sequences of the three strains were also deposited into the NCBI GenBank database under the following accession numbers: AmX2, JAFIWC000000000 (BioProject no. PRJNA700118); FX1, JAFJNT000000000 (BioProject no. PRJNA700116); and FX4, JAFIWB000000000 (BioProject no. PRJNA700117). Furthermore, pure cultures of the strains have been deposited in the CIRM-CFBP culture collection (French Collection for Plant-associated Bacteria) and assigned the following accession numbers: AmX2, CFBP 8902; FX1, CFBP 8700, FX2, CFBP 8701, FX3, CFBP 8702, and FX4, CFBP 8703.

## SUPPLEMENTAL MATERIAL

Supplemental material is available online only.

**SUPPLEMENTAL FILE 1**, PDF file, 4.9 MB.

## ACKNOWLEDGMENTS

We thank Cécile Dutrieux and Audrey Lathus at CIRM-CFBP for preservation of the bacterial strains.

E.O. conceived and designed the study with assistance from H.M. and S.Z. H.M. and S.Z. carried out the experiments with assistance from S.M.T., M.S.H., N.K., and I.D. E.O. analyzed and interpreted the data with assistance from T.R., H.M., and R.K. Technical support was provided by P.P. and M.F.-L.S. E.O. prepared the paper with assistance from H.M. All the authors revised the final version of the manuscript.

This article is based upon work from COST Action CA16107 EuroXanth, supported by COST (European Cooperation in Science and Technology). The work of N.K. was funded by the Deutsche Forschungsgemeinschaft (DFG, German Research Foundation), Project number 429677233. I.D. acknowledges the Ministry of Education, Science and Technological Development of the Republic of Serbia and INTERREG ADRION OIS-AIR (Open Innovation System of the Adriatic-Ionian Region) proof-of-concept project for the financial support of his research (contracts 451-03-9/2021-14/200178 and OIS-AIR 77). H.M. and S.Z. benefited from a sabbatical grant provided by the Iranian Ministry of Science and Technology for a 6-month fellowship at the University of Tehran, Iran. The work of E.O. was funded by University of Tehran, Iran.

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
