## [Reviewer comments · Microbiology Spectrum]

Microbiology Spectrum

Phenotypic and Molecular-Phylogenetic Analyses Revealed Distinct Features of Crown Gall-Associated Xanthomonas Strains

Hamzeh Mafakheri, S. Mohsen Taghavi, Sadegh Zarei, Touraj Rahimi, Mohammad Sadegh Hasannezhad, Perrine Portier, Marion Le-Saux, Ivica Dimkić, Ralf Koebnik, Nemanja Kuzmanović, and Ebrahim Osdaghi

Corresponding Author(s): Ebrahim Osdaghi, University of Tehran

Review Timeline:

Submission Date:	August 17, 2021
Editorial Decision:	September 13, 2021
Revision Received:	December 1, 2021
Editorial Decision:	December 3, 2021
Revision Received:	December 7, 2021
Accepted:	December 8, 2021

Editor: Lindsey Burbank

Reviewer(s): Disclosure of reviewer identity is with reference to reviewer comments included in decision letter(s). The following individuals involved in review of your submission have agreed to reveal their identity: Michael O'Leary (Reviewer #1)

Transaction Report:

DOI: <https://doi.org/10.1128/spectrum.00577-21>

September 13, 2021

Dr. Ebrahim Osdaghi
University of Tehran
Department of Plant Protection
Department of Plant Protection
University of Tehran
Tehran
Iran

Re: Spectrum00577-21 (Phenotypic and Molecular-Phylogenetic Analyses Revealed Distinct Features of Crown Gall-Associated Xanthomonas Strains)

Dear Dr. Ebrahim Osdaghi:

Thank you for submitting your manuscript to Microbiology Spectrum. The topic is relevant to the Spectrum audience, however there are still some issues that need to be addressed. The reviewers have provided detailed and constructive evaluations of the manuscript and analyses performed in your study. I would encourage you to carefully take these comments into account to improve your manuscript and to make sure that all conclusions are fully supported. In particular there are a number of points brought up in the previous review that have not been adequately addressed, and limitations that need to be discussed for some of the experiments presented. In addition the reviewers have made a number of suggestions to further clarify the writing and make it easier for the reader to interpret. Reviewers comments are provided at the bottom of this email as well as the attached file.

When submitting the revised version of your paper, please provide (1) point-by-point responses to the issues raised by the reviewers as file type "Response to Reviewers," not in your cover letter, and (2) a PDF file that indicates the changes from the original submission (by highlighting or underlining the changes) as file type "Marked Up Manuscript - For Review Only". Please use this link to submit your revised manuscript - we strongly recommend that you submit your paper within the next 60 days or reach out to me. Detailed information on submitting your revised paper are below.

Link Not Available

Sincerely,

Lindsey Burbank

Journals Department
Reviewer comments:

Reviewer #2 (Comments for the Author):

In the manuscript entitled "Phenotypic and Molecular-Phylogenetic Analyses Revealed Distinct Features of Crown Gall-3 Associated Xanthomonas Strains", Mafakheri et al. described that they isolated several bacterial strains belonging to the genus Xanthomonas and found that these strains are associated with crown gall disease of plants. The authors used MLSA method to classified the phylogenetic relationships of the strains against the Xanthomonas spp, and revealed that FX1-FX4 are close

relatives of *X. translucens*, whereas Amx2 is in a special clade. Furthermore, they compared the genomes and gene contents of these strains with the representative bacterial species of *Xanthomonas*, which showed that the crown gall-associated strains have different gene compositions in virulence-associated genes.

Although the data reported in the manuscript is interesting to the readers of plant pathology, especially those study crown gall diseases, the work carried out preliminary. The first complete genome sequence and comparative genomic analysis of *Xanthomonas* have been reported for nearly 20 years, and there is numerous sequence data of the genus deposited in the public databases. Bacterial strains isolated in this work are very special because the role of *Xanthomonas* spp. in crown gall disease was overlooked. However, descriptive and comparative analyses failed to address some of the important questions, such as:

1. Is *Xanthomonas* spp. alone or the cooperation of these strains with agrobacterium caused the crown gall?
2. How about the gene function in regulating virulence of development of the crown gall, as comparative genomic analysis predicted?

Reviewer #3 (Comments for the Author):

1. Reviewer's comment: Authors have isolated *Xanthomonas* strains from crown-gall samples, conducted Koch's postulates, and analyzed their genomes. Given the previous studies indicating that absence of T3SS and effectors in xanthomonads belong to clade I isolated from wide variety of hosts, authors need to justify how this study advances our understanding of these non-pathogenic xanthomonads. How are your findings relevant to plant-microbe interaction studies? Authors can improve this section rather than repeating what is already stated in the abstract. -

Author's response:

As stated in the "Importance" section of the manuscript, the microclimate of crown gall and their accompanying microorganisms has rarely been studied for the microbial diversity and population dynamics of gall-associated bacteria. In this study, we highlight the phenotypic characteristics and genomic features of gall associated xanthomonads. Our analyses not only revealed that biological behaviors and compatibility of *Xanthomonas* strains in the presence of their agrobacterial counterpart, they also shed light on the phylogenetic position, taxonomic status and virulence-associated genomic feature of the strains.

Reviewer's reply: What do you mean by compatibility? Have authors really addressed biological behavior of the *Xanthomonads* in presence of agrobacterial counterparts? Have they looked at population dynamic of gall associated xanthomonads in presence of *Agrobacterium* on Amaranth? The manuscript starts with crown gall associated *Xanthomonads*, but then focus suddenly shifts to the clade I xanthomonads and diversity within clade I. It seems that somehow overall question of ecology of these xanthomonads in presence of *Agrobacterium* is not addressed.

2. Reviewer's comment: Fig 1D-F- if you observed water-soaking phenotype, it is clearly not hypersensitive response! Authors have this completely wrong interpreted. Why was tomato not included in the population? Why was wheat phenotype not shown? Based on phylogenetic placement of these strains, one would expect to have these strains tested on hosts on which closely related strains are pathogens, eg. Grains, tea, etc.

Author's response: Fig 1D-F (Lines 167-169): The sentences have been changed as follows: In the leaf infiltration method, hypersensitivity reaction i.e. necrotic areas with a blackish-brown appearance was observed in the site of inoculation on squash (Figure 1D) melon (Figure 1E), and tomato (Figure 1F) as well as tobacco and geranium leaves 36-48 hpi. - Please note that we have inoculated wheat plants with both spray inoculation and injection techniques. Since we did not observe any symptoms on wheat plants inoculated with the bacterial strains, it does not make sense to include a picture of asymptomatic wheat plant.

Reviewer's reply: If there is hypersensitivity reaction and based on your observation of T3SS presence/absence, do you think these strains get recognized by different plant species? If not, how are these strains able to maintain their populations on the non-hosts?

Reviewer's comment: Any other better way to represent fig 1G to J? It is hard to compare populations over time when plotted as separate graphs. Why not arrange growth curves by host? It is interesting to see populations on certain hosts do not change over 28-day period. Why? Answering above questions might be useful to interpret about non-pathogenic nature of these strains.

Author's response: It could be possible to present the population dynamics on each plant species as a separate graph. However, in that case it would be very hard to compare different plant species with one another in the course of time.

Reviewer's reply: Refer to my above comment again. What is the rationale to compare populations among different plant species at a given sampling time? Isn't it easier to interpret on strain's ability to sustain populations on a given host by following growth curve/survival curve over time? It is interesting that wheat showed higher populations, yet it was asymptomatic among all hosts tested.

Reviewer's comment: There are several analyzes conducted in the comparative genomics section. For example, bacteriocins, phage, genomic islands. Authors might want to state the rationale for conducting these analyses. What is your hypothesis? In absence of a hypothesis, this laundry list of analyzes make no sense.

Author's response: These analyses altogether provide a comprehensive insight into the genome arrangement, virulence repertoires, and potential biological capabilities of the new xanthomonad clades. Since these strains represent a new set of taxa in terms of phylogenetic position and taxonomic status such analyses seem to help better understanding of the genetic diversity within the clade I of xanthomonads.

Reviewer's reply: Authors have convinced us on the diversity within clade I. However, how does fig 4A data help you to infer on ecology of Xanthomonads co-occurring with Agrobacterium?

Overall manuscript writing needs significant amount of work. I miss overall rationale behind this study. I see several pieces that authors are trying to put together, such as comparative genomics, pathogenicity tests, new species proposition etc. But what does it mean to simultaneously isolate these strains from crown gall samples? Is studying genomes to explain ecology of these strains one of the goals?

Author's response: This is a preliminary study to identify, characterize and describe the new atypical xanthomonad strains isolated from crown gall tissues of plants simultaneously infected with Agrobacterium strains. The phenotypic and genotypic data provided in this study along with the whole genome sequence resources will further pave the way of research on the clade I of xanthomonads. This study follows our previous projects i.e. <https://journals.asm.org/doi/full/10.1128/AEM.01518-19> , <https://apsjournals.apsnet.org/doi/10.1094/PHYTO-11-19-0428-R> , and <https://journals.asm.org/doi/full/10.1128/AEM.01518-19> on the clade I of xanthomonads. Further genomic investigations are undergoing to shed a light on the evolutionary relationships of the strains isolated in this study with those infecting small grain cereals.

Reviewer's reply: Authors have been engaged with significant research on clade I xanthomonads. Overall, the manuscript falls short of connecting the dots. If exploring clade I diversity is the goal, authors have convincing data. But if characterizing xanthomonads simultaneously infecting along with Agrobacterium, there is more to infer from the analyses already conducted in this manuscript.

Staff Comments:

Preparing Revision Guidelines

Please return the manuscript within 60 days; if you cannot complete the modification within this time period, please contact me. If

you do not wish to modify the manuscript and prefer to submit it to another journal, please notify me of your decision immediately so that the manuscript may be formally withdrawn from consideration by Microbiology Spectrum.

This manuscript describes the isolation and characterization of five *Xanthomonas* strains from naturally occurring *Agrobacterium*-induced galls on weeping fig, and the subsequent sequencing, assembly, and analysis of draft genomes for three of these strains. Although seemingly non-pathogenic *Xanthomonas* strains can be recovered from a variety of plant material and debris, comparatively little work has been done to characterize such strains relative to pathogenic isolates, as evidenced by the isolation of two potentially new species in this manuscript. However, there are several aspects that could be improved to strengthen the overall manuscript.

Regarding the text, there are many places where the narrative is difficult to follow, is repetitive with previous sections, or seems out of place (e.g., context and justification for experiments sometimes in Methods rather than Results, text appropriate for legends in Results, etc.). Explicitly justifying experiments and bioinformatic analyses, as well as providing context for the results, will help the reader understand the interesting work the authors have done and follow the narrative they have presented. The bioinformatics analyses would be strengthened by providing specific detail about parameters used as well as running quality control assessments to estimate assembly completeness (a critical assumption to verify for the comparative genomics section). Below, I have provided specific suggestions and comments:

Line 186. Need to add reference for these primers.

Line 201-202. "...indicate possible recombination". This sentence seems more appropriate for the figure 2 legend.

Line 209-211. It would be useful for the reader to add a comment regarding if these genome sizes and GC% are typical for *Xanthomonas* Clade I strains.

Line 222-227. Please clarify which of the three methods was used to generate the data shown in Table 2, currently it is not clear. It may also be helpful to spell out the full name of the abbreviations ANI and dDDH the first time they are used. Add ANI and dDDH values for species threshold here for reader comprehension. It would be helpful here to explicitly state the significant finding that AmX2 is below the ANI threshold for species identity with FX1/FX4 – referring to 'other *Xanthomonas* species' is ambiguous and could refer to the reference strains used rather than FX1/FX4. Finally, it would be helpful to give the reader an idea of how comprehensive the reference strains included in the ANI analysis were and how many in total were included out of available data – were all clade-I strains used...?

Line 233. Missing reference to figure 3A?

Line 234. “...T3SS repertoires...” It would be useful to briefly reintroduce the T3SS, T3E’s, the rationale for looking at the T3SS/T3Es, the implications of lacking a T3SS, and the lack of a T3SS in FX1/FX4/AMX2 before pointing the reader to Figure 3B.

Line 237. “...high query coverage but low sequence similarity...” Is it possible that these are flagellar proteins?

Line 242. I suggest introducing a paragraph break at “As for the....”.

Line 253. “Circular mapping...” Can the authors clarify if this was sequence read mapping? If so, explicitly stating that mapping sequencing reads against reference genomes provides additional support that sequences appearing in that reference genome are truly absent (or present) in the strain from which reads were generated (rather than simply being unassembled) would be helpful to provide context for this analysis to the reader. That said, while it is clear for some T3Es, it’s very difficult to see that T3SS structural genes are absent based on Figure S1. This figure would be more compelling if it zoomed in on that region, or maybe if read mapping depth vs. average genome mapping depth was provided for these genes from the sequenced strains.

Line 271-276. “One-vs-all collinearity test...” This analysis needs some work, or possibly removal. My interpretation of figure 4B is that the authors have used Mauve to re-order contigs against the strain DSM 18974 reference genome prior to alignment, which should be noted but is absent from this description as well as the methods (Line 541-542. “...Mauve software was used...”). Additionally, I would caution the authors against drawing conclusions about chromosomal structural variation based on re-ordered contigs from draft assemblies. Mauve does not take into account any potentially known spatial relationship data between contigs (i.e., from an assembly graph), only generates a ‘best-fit’ alignment of the draft to the reference. Fundamentally, alignments like this are similar to reference-guided assemblies in that the output for each strain will resemble the reference as much as possible, which is not necessarily the ‘ground truth’ for a novel strain. While it is entirely possible that any LCB identified which spans multiple contigs in one strain but not others does indeed have that arrangement and the contig break has been introduced by a repetitive sequence insertion, it is also entirely possible that a rearrangement has taken place and those contigs are not adjacent. The teal contig that spans 3000000 on DSM 18974 is an example of this situation, where a contig break is present in FX1 but not the other strains. A second example is the purple contig spanning 2000000 on DSM 18974, which is adjacent to a green contig in FX1 and FX4, but separated from the green contig in FX1 by a contig break – while this alignment looks like FX1 and FX4 have a shared rearrangement relative to DSM 18974, the contig break suggests that FX1 and FX4 may not be collinear in this region. It does not help that the contig breaks are difficult to distinguish from LCB connecting lines in this case. Sequence read mapping, PCR amplification, or long-read sequencing could potentially validate any rearrangements shown here, depending on the size of the gap between contigs. Unfortunately, based on the evidence currently presented, I cannot draw meaningful conclusions regarding collinearity or rearrangements among these

strains, only that the new strains seem like relatively complete assemblies relative to DSM 18974. I suggest the authors provide additional evidence for the LCB organization results mentioned in the text, or remove this particular analysis.

Line 277-281. "Three strains sequenced in this study...". Genes shared exclusively between FX1 and FX4 are potentially very interesting given their shared habitat. It would be valuable to indicate how many of these exist, and any potential annotations.

Line 294. "prophage sequences.... that were missing...". It's worthwhile to note that because these are draft genomes, it is possible that more phage sequences (which are often repetitive) are present in these strains but were not identified in the present study due to unresolved repetitive sequence.

Line 295. "Visualization of genomic islands...." This would benefit from additional explanation, both about genomic islands as well as the results. Did any of these islands contain interesting content, or is there evidence of horizontal gene transfer, or is there an island shared among strains....? Currently this feels tacked-on with no context for the reader.

Line 300-302. "genomic repertoires of five...strains...". I recommend rephrasing this section to be clear that five strains were isolated and phenotypically characterized, but only three were sequenced and compared.

Lines 322-330. Some of these sentences seem out of order or repetitive.

Line 357-359. "It could also be attributed to...". Were opine catabolism genes identified in these strains, maybe homologs of ooxAB? Can they grow on opines as a sole carbon source? I recommend rephrasing this sentence so it is clear that these strains if these strains have known opine catabolism genes or not.

Line 397-398. " non-pathogenic *Pseudomonas* strains..." these strains also typically fail to induce HR on tobacco, unlike the *Xanthomonas* strains in this study - a notable difference.

Lines 398-403. "An in-depth comparative analysis....several lines of evidence...". This is very vague. This section would be much stronger if the authors explicitly discussed the lines of evidence and precise insight gained.

Lines 436-439. I recommend moving the HR information to the Plant Inoculation Assay section beginning at line 447.

Line 471. “...weight of each leaf sample was adjusted to 2 g”. Could this influence the results in Figure 1 for comparisons between hosts? For example, by including far more inoculated leaves from one species by virtue of lower leaf mass and thus more CFUs.

Lines 463-4469. “Further, in order to...”. This is an example of information that would help the reader follow the narrative in the Results section (Lines 174-178), and is unnecessarily duplicated here

Line 502-509. Please provide more information regarding genome assembly, specifically any relevant non-default parameters used for bbdutk, trimmomatic, SPAdes (e.g., --careful, --isolate), which SPAdes pipeline was run, if any steps were taken to correct indels present in the assembly, etc. I also strongly recommend that the authors run at least one quality control analysis to assess the completeness and potential contamination in these genome assemblies. CheckM, BUSCO, or QUAST (using a high-quality clade I *Xanthomonas* reference genome) would be appropriate. A high completeness score with these methods would lend additional support to every genomics analysis in this manuscript, especially the absence of a T3SS and T3Es as a true feature of these genomes rather than potentially unassembled sequence.

Line 548. Were results from PlasmidFinder presented in this manuscript? I could not find them.

Lines 557-566, and 578-579. Delete duplicated section and add BioProject information, SRA information if submitted, etc.

Lines 806-815. Please indicate what measurement is represented by the error bars in Figure 1.

Line 840 (Figure 4C). What do the question marks (?) indicate on the OrthoVenn diagram?

Table 1. I recommend adding an additional score to clarify that the *Xanthomonas* sp. induced HR on Melon, Squash, etc. (e.g., ‘HR’, or ‘-*’ instead of ‘+’). The current presentation suggests that these strains are pathogenic on those hosts, whereas the evidence presented in lines 154-181 suggests that these strains are not pathogenic on the evaluated hosts.

Table 1, top row, change “Common been” to “Common bean”.

Figure 1. This figure would benefit from images of control treatments for hypersensitivity experiments e.g., sterile buffer, a *Xanthomonas* strain that does not induce HR on these hosts, and a *Xanthomonas* strain known to induce HR on these hosts. Additionally, providing a CFU estimate for the bacterial suspensions would help reproducibility. I agree with the previous reviewer #3 that 1G-1J can be presented more clearly. If I understand the author’s intention correctly, these plots should show that there is no meaningful bacterial population growth over time in these hosts, providing evidence that these strains are not pathogenic on these hosts, which is currently very difficult to see. For example, this data could be represented on two plots, one per strain, by clustering all four time points per host together on one axis, with different colors or patterns differentiating the time points. If a strain-vs-strain comparison is desired, one plot could be made using the same approach. Either of these presentations would focus on the time course aspect of this experiment and significantly improve the reader’s comprehension, while still allowing for a host-vs-host comparison.

Table S4. I recommend providing contig IDs, or gene IDs instead of start and end codons, for these proteins for genomes with more than one contig.

Table S5. Providing contig IDs would be useful here as well.

November 26, 2021

Dr. Lindsey Burbank
Editor
Microbiology Spectrum

Dear Dr. Burbank,

First of all, my co-authors and I would like to thank you and the three anonymous reviewers for considering our manuscript "Spectrum00577-21: Phenotypic and Molecular-Phylogenetic Analyses Revealed Distinct Features of Crown Gall-Associated *Xanthomonas* Strains" with a positive attitude. We deeply acknowledge that the reviewers have provided a number of critics, which markedly helped to increase the quality of this manuscript. We have addressed all the comments in the manuscript as precisely as possible. For your information all the changes made in the text as "track change mode" in an additional file named "Marked-Up Manuscript".

Thanks again for your time and looking forward to receiving the final decision on the manuscript.

With my best regards
On behalf of the co-authors
Ebrahim Osdaghi

Reviewer comments:

Reviewer #1 (Comments for the Author):

- This manuscript describes the isolation and characterization of five *Xanthomonas* strains from naturally occurring *Agrobacterium*-induced galls on weeping fig, and the subsequent sequencing, assembly, and analysis of draft genomes for three of these strains. Although seemingly non-pathogenic *Xanthomonas* strains can be recovered from a variety of plant material and debris, comparatively little work has been done to characterize such strains relative to pathogenic isolates, as evidenced by the isolation of two potentially new species in this manuscript. However, there are several aspects that could be improved to strengthen the overall manuscript.
- Regarding the text, there are many places where the narrative is difficult to follow, is repetitive with previous sections, or seems out of place (e.g., context and justification for experiments sometimes in Methods rather than Results, text appropriate for legends in Results, etc.). Explicitly justifying experiments and bioinformatic analyses, as well as providing context for the results, will help the reader understand the interesting work the authors have done and follow the narrative they have presented.
- The bioinformatics analyses would be strengthened by providing specific detail about parameters used as well as running quality control assessments to estimate assembly completeness (a critical assumption to verify for the comparative genomics section). Below, I have provided specific suggestions and comments:
 - **Line 198.** Need to add reference for these primers.
 - The references for all primer sequences used in this study, their annealing temperatures and the corresponding nucleotide sequence are summarized in Table S1.
- **Line 211-212.** "...indicate possible recombination". This sentence seems more appropriate for the figure 2 legend.
 - The sentence has been transferred to the legend of Figure 2 as follows: Figure 2: Phylogeny of crown gall-associated *Xanthomonas* strains isolated in this study based on the concatenated

sequences of four housekeeping genes, i.e. *atpD*, *efp*, *gyrB*, and *rpoD* using tree-based Maximum Likelihood (A) and network-based NeighborNet (B) analyses. The five *Xanthomonas* strains isolated from weeping fig and *Amaranthus* sp. were divided into two clades where the weeping fig strains were closely related to but still distinct from the *X. translucens* species complex. The *Amaranthus* sp. strain was distinct from all the validly described species with the clade-I of *Xanthomonas*. Further, parallel lines/branches within the nodes of NeighborNet network indicate possible recombination between the strains being connected to each other.

- **Line 223-225.** It would be useful for the reader to add a comment regarding if these genome sizes and GC% are typical for *Xanthomonas* Clade I strains.

- The sentence has been changed as follows: Genome size of the strains was 5,383 kbp (in 152 contigs: FX1); 5,383 kbp (in 89 contigs: FX4), and 5,069 kbp (in 75 contigs: AmX2), with the GC% ranging from 68.8% to 69.8% which is within the typical range of clade I *Xanthomonas* strains.

- **Line 237-244.** Please clarify which of the three methods was used to generate the data shown in Table 2, currently it is not clear. It may also be helpful to spell out the full name of the abbreviations ANI and dDDH the first time they are used. Add ANI and dDDH values for species threshold here for reader comprehension. It would be helpful here to explicitly state the significant finding that AmX2 is below the ANI threshold for species identity with FX1/FX4 – referring to ‘other *Xanthomonas* species’ is ambiguous and could refer to the reference strains used rather than FX1/FX4. Finally, it would be helpful to give the reader an idea of how comprehensive the reference strains included in the ANI analysis were and how many in total were included out of available data – were all clade-I strains used...?

- The following sentence has been added to the text to further clarify the method: In order to estimate DNA similarity indexes between the strains sequenced in this study and all *Xanthomonas* species, ANI values were calculated using three online services JSpeciesWS, ANI calculator, and OrthoANIu where the ultimate ANI value was obtained via combination and averaging of the results generated by the three online services.

- The full name of the abbreviations ANI and dDDH have been spelled out in their first use in the text.

- The following sentence has been added to the text to for readers’ comprehension: These ANI and dDDH values are far below the accepted threshold for prokaryotic species definition, i.e. $\leq 95\%$ and $\leq 70\%$ for ANI and dDDH, respectively, suggesting that the strains FX1, FX4, and AmX2 belong to two new species within the clade-IA of *Xanthomonas*.

- The following sentence has been added to the text to highlight the taxonomic distinction of FX1, FX4, and AmX2: These ANI and dDDH values are far below the accepted threshold for prokaryotic species definition, i.e. $\leq 95\%$ and $\leq 70\%$ for ANI and dDDH, respectively, suggesting that the strains FX1, FX4, and AmX2 belong to two new species within the clade-IA of *Xanthomonas*. While the two strains FX1 and FX4 belong to the same species, the strain AmX2 could be considered as another standalone species both distinct from all validly-described *Xanthomonas* spp. (Figure 3A)..

- The following sentence has been added to the text to clarify how comprehensive the reference strains included in the ANI analysis were and how many in total were included out of available data: According to the MLSA results which confirmed taxonomic position of the strains isolated in this study in clade I of *Xanthomonas* spp., type strains of all clade I *Xanthomonas* species were included in DNA similarity analyses.

- **Line 254.** Missing reference to figure 3A?

- The reference for Figure 3A has been added to the text.

- **Line 258-263.** “...T3SS repertoires...” It would be useful to briefly reintroduce the T3SS, T3E’s, the rationale for looking at the T3SS/T3Es, the implications of lacking a T3SS, and the lack of a T3SS in FX1/FX4/AMX2 before pointing the reader to Figure 3B.

- A short description of the T3SS, T3E’s and TALEs is already existed in the introduction section.
- The following sentences have been added to the text to highlight the implications of lacking a T3SS, and the lack of a T3SS in FX1/FX4/AMX2: It has been shown that the clade I Xanthomonas species possess high genetic diversity in terms of T3SS, T3Es, and TALEs. Genome-informed investigation of Xanthomonas spp. would provide valuable information on the virulence repertoires, pathogenicity mechanisms, and host adaptation of the strains (9).

- **Line 268-273.** “...high query coverage but low sequence similarity...” Is it possible that these are flagellar proteins?

- The following sentences have been added to the text to further clarify the issue: For instance, a sequence similar to hrpB6 (94% query coverage and 48% sequence similarity) was predicted in AmX2, FX1, and FX4. The hrpB6 is a T3SS ATPase in *X. euvesicatoria* (GeneID: 61777304). Furthermore, hrpC2 (originally described in *X. euvesicatoria*; accession number P80150) which belongs to FHIPEP (flagella/HR/invasion proteins export pore) family was detected in all the three strains with 97% query coverage and 32% similarity.

- **Line 277.** I suggest introducing a paragraph break at “As for the....”.

- A paragraph break has been inserted in this position.

- **Line 287-289.** “Circular mapping...” Can the authors clarify if this was sequence read mapping? If so, explicitly stating that mapping sequencing reads against reference genomes provides additional support that sequences appearing in that reference genome are truly absent (or present) in the strain from which reads were generated (rather than simply being unassembled) would be helpful to provide context for this analysis to the reader. That said, while it is clear for some T3Es, it’s very difficult to see that T3SS structural genes are absent based on Figure S1. This figure would be more compelling if it zoomed in on that region, or maybe if read mapping depth vs. average genome mapping depth was provided for these genes from the sequenced strains.

- Please note that we used the FASTA file of the genome sequences obtained in this study for circular mapping. Furthermore, quality assessment of genome assembly and annotation using BUSCO online service confirmed the accuracy and completeness of all three genomes compared to the complete genome sequence of *X. translucens* DSM 18974 (**Line 219-223**). Hence, it could be hypnotized with a considerable confidence that the presence/absence scheme of the genes provided by circular mapping is reliable in the extent of this manuscript. Finally, BRIG software using which the pairwise genome collinearity alignment of the three strains was performed and visualized does not have any zoom in/out option so that the generated circular map could not be zoomed out. In conclusion, we believe that the dataset provided in these analyses are satisfactory enough to shed a light on the genomic contents and virulence-associated repertoires of the Xanthomonas strains isolated from crown gall tissues.

- **Line 306-311.** “One-vs-all collinearity test...” This analysis needs some work, or possibly removal. My interpretation of figure 4B is that the authors have used Mauve to re-order contigs against the strain DSM 18974 reference genome prior to alignment, which should be noted but is absent from this description as well as the methods (**Line 640-641.** “...Mauve software was used....”). Additionally, I

would caution the authors against drawing conclusions about chromosomal structural variation based on re-ordered contigs from draft assemblies. Mauve does not take into account any potentially known spatial relationship data between contigs (i.e., from an assembly graph), only generates a 'best-fit' alignment of the draft to the reference. Fundamentally, alignments like this are similar to reference guided assemblies in that the output for each strain will resemble the reference as much as possible, which is not necessarily the 'ground truth' for a novel strain. While it is entirely possible that any LCB identified which spans multiple contigs in one strain but not others does indeed have that arrangement and the contig break has been introduced by a repetitive sequence insertion, it is also entirely possible that a rearrangement has taken place and those contigs are not adjacent. The teal contig that spans 3000000 on DSM 18974 is an example of this situation, where a contig break is present in FX1 but not the other strains. A second example is the purple contig spanning 2000000 on DSM 18974, which is adjacent to a green contig in FX1 and FX4, but separated from the green contig in FX1 by a contig break – while this alignment looks like FX1 and FX4 have a shared rearrangement relative to DSM 18974, the contig break suggests that FX1 and FX4 may not be colinear in this region. It does not help that the contig breaks are difficult to distinguish from LCB connecting lines in this case. Sequence read mapping, PCR amplification, or long-read sequencing could potentially validate any rearrangements shown here, depending on the size of the gap between contigs. Unfortunately, based on the evidence currently presented, I cannot draw meaningful conclusions regarding collinearity or rearrangements among these strains, only that the new strains seem like relatively complete assemblies relative to DSM 18974. I suggest the authors provide additional evidence for the LCB organization results mentioned in the text, or remove this particular analysis.

- We do agree with the explanations provided by the reviewer, and admit that the sequence data provided in this study are not appropriate to be analyzed with Mauve. The co-linearity analyses which were performed by Mauve software to illustrate locally collinear blocks among the same set of genomes were removed from the manuscript. Figure 4 has been changed accordingly.

- **Line 311-327.** "Three strains sequenced in this study....". Genes shared exclusively between FX1 and FX4 are potentially very interesting given their shared habitat. It would be valuable to indicate how many of these exist, and any potential annotations.

- The text has been changed as follows: Orthologous gene clusters were determined using OrthoVenn online service through four-vs.-four designations of the strains where one of the clade-I strains DSM 18974 and CFBP 4691 were used as reference genome in each analyses (Figure 4B, C). When *X. translucens* strain DSM 18974 was considered as reference, the three strains sequenced in this study shared 611 proteins besides the 2,609 proteins which were shared among all the four strains. The strains AmX2, FX1, and FX4 possessed 35, four and three unique proteins, respectively when DSM 18974 was designated as reference genome. Furthermore, FX1 and FX4 shared 584 proteins with each other while these strains shared respectively 29 and 35 proteins with AmX2 indicating closer genomic repertoires of FX1 and FX4. On the other hand, when *X. theicola* CFBP 4691 was designated as reference genome the three strains sequenced in this study showed 811 shared proteins along with 2,416 proteins which were shared within all the four genomes. The strains AmX2, FX1, and FX4 possessed 37, three and three unique proteins respectively, when CFBP 4691 was considered as reference genome (Figure 4B, C).

- **Line 341-344.** "prophage sequences.... that were missing...". It's worthwhile to note that because these are draft genomes, it is possible that more phage sequences (which are often repetitive) are present in these strains but were not identified in the present study due to unresolved repetitive sequence.

- The following sentence has been added to the text: However, it should be noted that due to the draft nature of the genomes, it is possible that more phage sequences are present in these strains but were not identified in the present study due to unresolved repetitive sequence.

Line 345-360. “Visualization of genomic islands....” This would benefit from additional explanation, both about genomic islands as well as the results. Did any of these islands contain interesting content, or is there evidence of horizontal gene transfer, or is there an island shared among strains....? Currently this feels tacked-on with no context for the reader.

- The following explanation has been added to the text to clarify the status of genomic islands in the strains sequenced in this study: Visualization of genomic islands in the whole genome sequences of the three *Xanthomonas* sp. strains obtained in this study against the genomes of two reference strains *X. theicola* CFBP 4691 and *X. translucens* pv. *translucens* DSM 18974 is shown in Figure S2. Distribution pattern of the main genomic islands in all the three strains sequenced in this study remained constant regardless of the reference genome used for prediction. When the strain CFBP 4691 was considered as reference genome, two dominant genomic islands were predicted in FX1 positioning between 0.9M-1.0M and 2.5M-2.6M. The two islands were predicted when DSM 18974 was used as reference although the position of the islands was slightly different (1.5M-1.6M and 2.8M-2.9M). Similarly, a genomic island was constantly predicted in FX4 when both the reference genomes were used, while position of the island has changed from 2.1M-2.2M to 2.9M-3.0M when CFBP 4691 and DSM 18974 were used as reference genome, respectively (Figure S2). The same pattern was observed in the strain AmX2 where a dominant genomic island was predicted on 1.6M-1.7 and 2.3M-2.4M when CFBP 4691 and DSM 18974 were used as reference genome, respectively. These results indicate different patterns of genomic islands distribution among the strains sequenced in this study despite the fact that they belong to the same species, i.e. FX1 and FX4 (Figure S2).

Line 364-372. “genomic repertoires of five...strains...”. I recommend rephrasing this section to be clear that five strains were isolated and phenotypically characterized, but only three were sequenced and compared.

- The text has been changed as follows: Five *Xanthomonas* sp. strains have accidentally been isolated from crown gall tissues along with several *Agrobacterium* strains, while the latter organisms were the main causal agent of the disease. In this study, we employed a series of biochemical tests, pathogenicity assays and MLSA-based phylogenetic analyses on five crown gall-associated *Xanthomonas* strains isolated from weeping fig and *Amaranthus* sp. plants in Iran. Subsequently, three representative strains, i.e. FX1, FX4 and Amx2 were subjected to genome sequencing and comparative genomics to elucidate their biological features, taxonomic position and genomic repertoires. The strains investigated in this study induced hypersensitive reaction (HR) on geranium, melon, squash, tobacco and tomato leaves while they were non-pathogenic on their host of isolation. MLSA-based phylogenetic analyses accomplished with whole genome sequence-based ANI/dDDH calculations suggested that the *Xanthomonas* strains isolated from crown gall tissues could be considered as two novel species within the clade-I of *Xanthomonas*.

Lines 396-404. Some of these sentences seem out of order or repetitive.

- The text has been re-phrased as follows: A physiologically active crown gall produces specific nutrients (opines) utilizable by microorganisms providing a rich niche for plant-associated bacteria to colonize, grow, and form complex ecological interactions (19, 20; 21; 22). The microbial communities involved in opine ecology is not limited to the *Agrobacterium* members.

Complex bacterial communities of many species co-exist in crown gall tissues (23). During the past few decades, various bacterial species, i.e. *Pseudomonas* spp. and coryneform bacteria were isolated from crown gall tissues possessing the capability of competing for opines as growth substrates (19).

- **Line 331-337.** “It could also be attributed to...”. Were opine catabolism genes identified in these strains, maybe homologs of *ooxAB*? Can they grow on opines as a sole carbon source? I recommend rephrasing this sentence so it is clear that these strains if these strains have known opine catabolism genes or not.

- The following sentences have been added to the text: BLAST-based in-silico analyses using the sequences of *ooxA*, *ooxB*, and *ooxAB* genes against the three genome sequences obtained in this study revealed the presence of these genes in the three *Xanthomonas* strains. As detailed in Table S2B, sequences of DNA fragments with 66-98% query coverage and 25-27% sequence similarity to *ooxA*, *ooxB*, and *ooxAB* genes were detected in the genome of all three strains. These evidences suggest the potential acquisition of opine catabolism genes by gall-associated xanthomonads which needs further in-depth investigations.

- **Line 478-482.** “non-pathogenic *Pseudomonas* strains...” these strains also typically fail to induce HR on tobacco, unlike the *Xanthomonas* strains in this study - a notable difference.

- The following sentence has been added to the text: These non-pathogenic *Pseudomonas* strains also typically fail to induce HR on tobacco, unlike the *Xanthomonas* strains isolated in this study.

- **Lines 482-494.** “An in-depth comparative analysis....several lines of evidence...”. This is very vague. This section would be much stronger if the authors explicitly discussed the lines of evidence and precise insight gained.

- The following sentences have been added to the text: Subsystem analyses using RAST service highlighted differences in the genetic contents of clade-I xanthomonads especially when it comes to the differences between the cereal pathogen *X. translucens* and the strains isolated in this study. For instance, the number of subsystems that are responsible for biotin biosynthesis, invasion and intracellular resistance, siderophore production, protein biosynthesis, nitrogen metabolism, central aromatic intermediates metabolism, sulfur metabolism, methylglyoxal metabolism, potassium metabolism, maltose and maltodextrin utilization, respiration and the miscellaneous subsystems in the strains FX1 and FX4 was higher than those in the members of their neighbor clade *X. translucens*. Furthermore, prophage sequences were missing in the strains sequenced in this study, but since these are draft genomes, it is possible that more phage sequences (which are often repetitive) are present in these strains but were not identified in the present study due to unresolved repetitive sequence.

- **Lines 538-540.** I recommend moving the HR information to the Plant Inoculation Assay section beginning at line 538.

- Done.

- **Line 565.** “...weight of each leaf sample was adjusted to 2 g”. Could this influence the results in Figure 1 for comparisons between hosts? For example, by including far more inoculated leaves from one species by virtue of lower leaf mass and thus more CFUs.

- We do agree with the reviewer’s opinion that using similar weight of all plant species could possibly influence the final data. However, if one wants to use different weights of leaf samples in different plant species, the resulting data would be incomparable, and it would be almost

impossible to summarize and present the data in one graph. To create uniform experimental conditions in this experiment, we took the same weight from the leaves.

- **Lines 559-561.** “Further, in order to...”. This is an example of information that would help the reader follow the narrative in the Results section (**Lines 178-180**), and is unnecessarily duplicated here

- The following sentences have been moved to the Results section: Furthermore, in order to provide a precise insight into the biology of the strains isolated in this study, in planta growth of the strains AmX2 and FX1 was investigated on 12 plant species, i.e. Amaranthus sp., barley, black nightshade (*Solanum nigrum*), red nightshade (*Solanum villosum*), common bean, common zinnia (*Zinnia elegans*), maize, melon, spinach (*Spinacia oleracea* L. cv. hudson), squash, watermelon, and wheat under greenhouse conditions. Population size of the inoculated bacterial strains were counted 7, 14, 21, and 28 dpi and presented in log CFU/g of the leaf tissue (Figure 1I, J).

- **Line 599-603.** Please provide more information regarding genome assembly, specifically any relevant non-default parameters used for bbduk, trimmomatic, SPAdes (e.g., --careful, --isolate), which SPAdes pipeline was run, if any steps were taken to correct indels present in the assembly, etc. I also strongly recommend that the authors run at least one quality control analysis to assess the completeness and potential contamination in these genome assemblies. CheckM, BUSCO, or QUAST (using a high-quality clade I *Xanthomonas* reference genome) would be appropriate. A high completeness score with these methods would lend additional support to every genomics analysis in this manuscript, especially the absence of a T3SS and T3Es as a true feature of these genomes rather than potentially unassembled sequence.

- Quantitative assessment of genome assembly and annotation completeness was evaluated using BUSCO online software.

- The following sentences have been added to the text: The read quality was assessed with FastQC (Galaxy version 0.72+galaxy1; <http://www.bioinformatics.babraham.ac.uk/projects/fastqc/>), while quantitative assessment of genome assembly and annotation completeness was evaluated using BUSCO online software (46; 47).

- **Line 343-344.** Were results from PlasmidFinder presented in this manuscript? I could not find them.

- The following sentence is the result of PlasmidFinder analyses: No integrative plasmid (episome) was detected using the PlasmidFinder online service in the genome of strains in this study.

- **Lines 660-662, and 676-677.** Delete duplicated section and add BioProject information, SRA information if submitted, etc.

- Done.

- BioProject data of the strains have been added to the text as follows: Whole genome sequences of the three strains were also deposited into the NCBI GenBank database under accession numbers for AmX2: JAFIWC000000000 (BioProject: PRJNA700118); FX1: JAFJNT000000000 (BioProject: PRJNA700116); and FX4: JAFIWB000000000 (BioProject: PRJNA700117).

- **Lines 917-918.** Please indicate what measurement is represented by the error bars in Figure 1.

- The following sentence has been added to the legend of Figure 1: Error bars indicate standard deviation.

- **Line 944-953** (Figure 4C). What do the question marks (“?”) indicate on the OrthoVenn diagram?

- Figure 4 has been redrawn completely.

- Table 1. I recommend adding an additional score to clarify that the *Xanthomonas* sp. induced HR on Melon, Squash, etc. (e.g., 'HR', or '-*' instead of '+'). The current presentation suggests that these strains are pathogenic on those hosts, whereas the evidence presented in lines 154-181 suggests that these strains are not pathogenic on the evaluated hosts.

- Table 1 has been corrected accordingly.

- Table 1, top row, change "Common been" to "Common bean".

- Corrected.

- Figure 1, This figure would benefit from images of control treatments for hypersensitivity experiments e.g., sterile buffer, a *Xanthomonas* strain that does not induce HR on these hosts, and a *Xanthomonas* strain known to induce HR on these hosts. Additionally, providing a CFU estimate for the bacteria suspensions would help reproducibility. I agree with the previous reviewer #3 that 1G-1J can be presented more clearly. If I understand the author's intention correctly, these plots should show that there is no meaningful bacterial population growth over time in these hosts, providing evidence that these strains are not pathogenic on these hosts, which is currently very difficult to see. For example, this data could be represented on two plots, one per strain, by clustering all four time points per host together on one axis, with different colors or patterns differentiating the time points. If a strain-vs-strain comparison is desired, one plot could be made using the same approach. Either of these presentations would focus on the time course aspect of this experiment and significantly improve the reader's comprehension, while still allowing for a host-vs-host comparison.

- Figure 1 has thoroughly been revised and all requested items have been added.

- Table S5. I recommend providing contig IDs, or gene IDs instead of start and end codons, for these proteins for genomes with more than one contig.

- The gene IDs have been added to the table.

- Table S6. Providing contig IDs would be useful here as well.

- The gene IDs have been added to the table.

Reviewer comments:

Reviewer #2 (Comments for the Author):

- In the manuscript entitled "Phenotypic and Molecular-Phylogenetic Analyses Revealed Distinct Features of Crown Gall-Associated *Xanthomonas* Strains", Mafakheri et al. described that they isolated several bacterial strains belonging to the genus *Xanthomonas* and found that these strains are associated with crown gall disease of plants. The authors used MLSA method to classify the phylogenetic relationships of the strains against the *Xanthomonas* spp, and revealed that FX1-FX4 are close relatives of *X. translucens*, whereas Amx2 is in a special clade. Furthermore, they compared the genomes and gene contents of these strains with the representative bacterial species of *Xanthomonas*, which showed that the crown gall-associated strains have different gene compositions in virulence-associated genes.

- Although the data reported in the manuscript is interesting to the readers of plant pathology, especially those study crown gall diseases, the work carried out preliminary. The first complete genome sequence and comparative genomic analysis of *Xanthomonas* have been reported for nearly 20 years, and there is numerous sequence data of the genus deposited in the public databases. Bacterial strains isolated in this work are very special because the role of *Xanthomonas* spp. in crown gall disease was overlooked.

However, descriptive and comparative analyses failed to address some of the important questions, such as:

- Is *Xanthomonas* spp. alone or the cooperation of these strains with agrobacterium caused the crown gall?
 - As indicated in the manuscript, the *Xanthomonas* strains isolated in this study did not induce any symptoms on their host of isolation nor on the other evaluated test plants except for HR which was observed on some annual crops. On the other hand, the accompanying *Agrobacterium* strains isolated from the same gall tissues caused crown gall symptoms on the test plants either in presence or absence of the *Xanthomonas* strains. Hence, the gall formation is a characteristic of *Agrobacterium* strains that is not affected by *Xanthomonas* strains. This issue is discussed in the manuscript. Furthermore, we have evaluated the opine-catabolic operon named *ooxAB* in the *Xanthomonas* strains isolated from gall tissue. The results are described in **lines 431-436**.

- How about the gene function in regulating virulence of development of the crown gall, as comparative genomic analysis predicted?
 - As explained in the previous comment, the *Xanthomonas* strains isolated in this study did not induce any symptoms on their host of isolation nor on the other evaluated test plants. Furthermore, they did not affect the gall formation function of their accompanying *Agrobacterium* strains. We do not claim any correlation with the genomic content of the *Xanthomonas* strains and formation of crown gall in the host plants. Comparative genomics of the *Xanthomonas* strains was performed to shed a light on the genetic contents of the strains in comparison with the other *Xanthomonas* species but not necessarily to find a correlation between their genomic content and the gall formation. Furthermore, we have evaluated the octopine-catabolic operon named *ooxAB* in the *Xanthomonas* strains isolated from gall tissue. Evidences of the presence of these genes in the gall-associated *Xanthomonas* strains have been described in **lines 431-436**.

Reviewer #3 (Comments for the Author):

- *Reviewer's comment:* Authors have isolated *Xanthomonas* strains from crown-gall samples, conducted Koch's postulates, and analyzed their genomes. Given the previous studies indicating that absence of T3SS and effectors in xanthomonads belong to clade I isolated from wide variety of hosts, authors need to justify how this study advances our understanding of these non-pathogenic xanthomonads. How are your findings relevant to plant-microbe interaction studies? Authors can improve this section rather than repeating what is already stated in the abstract. -
- *Author's response:* As stated in the "Importance" section of the manuscript, the microclimate of crown gall and their accompanying microorganisms has rarely been studied for the microbial diversity and population dynamics of gall-associated bacteria. In this study, we highlight the phenotypic characteristics and genomic features of gall associated xanthomonads. Our analyses not only revealed that biological behaviors and compatibility of *Xanthomonas* strains in the presence of their agrobacterial counterpart, they also shed light on the phylogenetic position, taxonomic status and virulence-associated genomic feature of the strains.
- *Reviewer's reply:* What do you mean by compatibility? Have authors really addressed biological behavior of the *Xanthomonads* in presence of agrobacterial counterparts? Have they looked at population dynamic of gall associated xanthomonads in presence of *Agrobacterium* on Amaranth? The manuscript starts with crown gall associated *Xanthomonads*, but then focus suddenly shifts to the clade I xanthomonads and diversity within clade I. It seems that somehow overall question of ecology of these xanthomonads in presence of *Agrobacterium* is not addressed.

- As we stated in the manuscript a couple of greenhouse experiments highlighted the biological properties of the *Xanthomonas* strains. First, we have evaluated the pathogenicity and host range of the strains not only on their host of isolation but also on some other annual crops using the *Xanthomonas* strains alone or in combination with their accompanying *Agrobacterium* strains. Second, we have evaluated the epiphytic growth and survival of the *Xanthomonas* strains on a series of annual crops which provides an insight into the extent of the possibility of their survival on the plant species other than their host of isolation.
- Furthermore, we have added a number of new analyses i.e. the opine catabolism genes in the *Xanthomonas* strains. We have evaluated the octopine-catabolic operon named *ooxAB* in the *Xanthomonas* strains isolated from gall tissue. The results are described in **lines 431-436**.

- *Reviewer's comment:* Fig 1D-F- if you observed water-soaking phenotype, it is clearly not hypersensitive response! Authors have this completely wrong interpreted. Why was tomato not included in the population? Why was wheat phenotype not shown? Based on phylogenetic placement of these strains, one would expect to have these strains tested on hosts on which closely related strains are pathogens, eg. Grains, tea, etc.

- *Author's response:* Fig 1D-F (**Lines 165-171**): The sentences have been changed as follows: In the leaf infiltration method, hypersensitivity reaction i.e. necrotic areas with a blackish-brown appearance was observed in the site of inoculation on squash (Figure 1D) melon (Figure 1E), and tomato (Figure 1F) as well as tobacco and geranium leaves 36-48 hpi. - Please note that we have inoculated wheat plants with both spray inoculation and injection techniques. Since we did not observe any symptoms on wheat plants inoculated with the bacterial strains, it does not make sense to include a picture of asymptomatic wheat plant.

- *Reviewer's reply:* If there is hypersensitivity reaction and based on your observation of T3SS presence/absence, do you think these strains get recognized by different plant species? If not, how are these strains able to maintain their populations on the non-hosts?

- Please note that the survival of plant pathogenic bacteria on non-host plant species is not necessarily controlled by their pathogenicity repertoires. Rather, there are several examples in the literature where it has been shown that both non-pathogenic and pathogenic variants of a given bacterial pathogen have almost equal epiphytic survival on non-host plant species. To see one of those examples please visit <https://bsppjournals.onlinelibrary.wiley.com/doi/full/10.1111/ppa.12730>

- *Reviewer's comment:* Any other better way to represent fig 1G to J? It is hard to compare populations over time when plotted as separate graphs. Why not arrange growth curves by host? It is interesting to see populations on certain hosts do not change over 28-day period. Why? Answering above questions might be useful to interpret about non-pathogenic nature of these strains.

- *Author's response:* It could be possible to present the population dynamics on each plant species as a separate graph. However, in that case it would be very hard to compare different plant species with one another in the course of time.

- *Reviewer's reply:* Refer to my above comment again. What is the rationale to compare populations among different plant species at a given sampling time? Isn't it easier to interpret on strain's ability to sustain populations on a given host by following growth curve/survival curve over time? It is interesting that wheat showed higher populations, yet it was asymptomatic among all hosts tested.

- Figure 1 has thoroughly been revised and all requested items have been added.

- *Reviewer's comment:* There are several analyzes conducted in the comparative genomics section. For example, bacteriocins, phage, genomic islands. Authors might want to state the rationale for conducting

these analyses. What is your hypothesis? In absence of a hypothesis, this laundry list of analyzes make no sense.

- *Author's response:* These analyses altogether provide a comprehensive insight into the genome arrangement, virulence repertoires, and potential biological capabilities of the new xanthomonad clades. Since these strains represent a new set of taxa in terms of phylogenetic position and taxonomic status such analyses seem to help better understanding of the genetic diversity within the clade I of xanthomonads.

- *Reviewer's reply:* Authors have convinced us on the diversity within clade I. However, how does fig 4A data help you to infer on ecology of Xanthomonads co-occurring with Agrobacterium?

- Please note that comparative genomics analyses using the data provided by RAST and SEED-Viewer databases highlighted differences in metabolic networks and subsystem profiles between the three strains sequenced in this study and those of the clade-I xanthomonads, as well as the reference strain *X. oryzae* pv. *oryzae* PXO99A. Results of these analyses will help the readers to find metabolic characteristics that differentiate the strains isolated in this study and those of plant pathogenic *Xanthomonas* spp.

- Overall manuscript writing needs significant amount of work. I miss overall rationale behind this study. I see several pieces that authors are trying to put together, such as comparative genomics, pathogenicity tests, new species proposition etc. But what does it mean to simultaneously isolate these strains from crown gall samples? Is studying genomes to explain ecology of these strains one of the goals?

- *Author's response:* This is a preliminary study to identify, characterize and describe the new atypical xanthomonad strains isolated from crown gall tissues of plants simultaneously infected with Agrobacterium strains. The phenotypic and genotypic data provided in this study along with the whole genome sequence resources will further pave the way of research on the clade I of xanthomonads. This study follows our previous projects i.e. <https://journals.asm.org/doi/full/10.1128/AEM.01518-19>, <https://apsjournals.apsnet.org/doi/10.1094/PHYTO-11-19-0428-R>, and <https://journals.asm.org/doi/full/10.1128/AEM.01518-19> on the clade I of xanthomonads. Further genomic investigations are undergoing to shed a light on the evolutionary relationships of the strains isolated in this study with those infecting small grain cereals.

- *Reviewer's reply:* Authors have been engaged with significant research on clade I xanthomonads. Overall, the manuscript falls short of connecting the dots. If exploring clade I diversity is the goal, authors have convincing data. But if characterizing xanthomonads simultaneously infecting along with Agrobacterium, there is more to infer from the analyses already conducted in this manuscript.

- The ultimate goal of this manuscript is to provide preliminary data on the biological characteristics and phylogenetic status of the gall-associated Xanthomonas strains and their taxonomic relationships with the clade I xanthomonads.

December 3, 2021

Dr. Ebrahim Osdaghi
University of Tehran
Department of Plant Protection
Department of Plant Protection
University of Tehran
Tehran
Iran

Re: Spectrum00577-21R1 (Phenotypic and Molecular-Phylogenetic Analyses Revealed Distinct Features of Crown Gall-Associated Xanthomonas Strains)

Dear Dr. Ebrahim Osdaghi:

Thank you for submitting your manuscript to Microbiology Spectrum. As you will see your paper is very close to acceptance. There are just a couple of points that still need to be addressed:

1. Please provide more details and parameters for the genome assemblies as suggested by reviewer #1. Even if this was done using a standardized pipeline such as Galaxy, the parameters should be described to facilitate reproducibility and use of your sequences by other researchers. The section I refer to is around line 600 in the materials and methods.
2. Please cite all software tools used when first mentioned in the text (BAGEL4, PHASTER, Plasmidfinder etc) so it is easier to find this information.

Please modify the manuscript along the lines I have recommended. As these revisions are quite minor, I expect that you should be able to turn in the revised paper in less than 30 days, if not sooner. If your manuscript was reviewed, you will find the reviewers' comments below.

When submitting the revised version of your paper, please provide (1) point-by-point responses to the issues I raised in your cover letter, and (2) a PDF file that indicates the changes from the original submission (by highlighting or underlining the changes) as file type "Marked Up Manuscript - For Review Only". Please use this link to submit your revised manuscript. Detailed instructions on submitting your revised paper are below.

Link Not Available

Sincerely,

Lindsey Burbank

Reviewer comments:

Preparing Revision Guidelines

- point-by-point responses to the issues I raised in your cover letter

- Upload a compare copy of the manuscript (without figures) as a "Marked-Up Manuscript" file.
- Each figure must be uploaded as a separate file, and any multipanel figures must be assembled into one file.
- Manuscript: A .DOC version of the revised manuscript
- Figures: Editable, high-resolution, individual figure files are required at revision, TIFF or EPS files are preferred

Please return the manuscript within 60 days; if you cannot complete the modification within this time period, please contact me. If you do not wish to modify the manuscript and prefer to submit it to another journal, please notify me of your decision immediately so that the manuscript may be formally withdrawn from consideration by Microbiology Spectrum.

November 26, 2021

Dr. Lindsey Burbank
Editor
Microbiology Spectrum

Dear Dr. Burbank,

First of all, my co-authors and I would like to thank you and the three anonymous reviewers for considering our manuscript "Spectrum00577-21: Phenotypic and Molecular-Phylogenetic Analyses Revealed Distinct Features of Crown Gall-Associated *Xanthomonas* Strains" with a positive attitude. We deeply acknowledge that the reviewers have provided a number of critics, which markedly helped to increase the quality of this manuscript. We have addressed all the comments in the manuscript as precisely as possible. For your information all the changes made in the text as "track change mode" in an additional file named "Marked-Up Manuscript".

Thanks again for your time and looking forward to receiving the final decision on the manuscript.

With my best regards
On behalf of the co-authors
Ebrahim Osdaghi

Reviewer comments:

Reviewer #1 (Comments for the Author):

- This manuscript describes the isolation and characterization of five *Xanthomonas* strains from naturally occurring *Agrobacterium*-induced galls on weeping fig, and the subsequent sequencing, assembly, and analysis of draft genomes for three of these strains. Although seemingly non-pathogenic *Xanthomonas* strains can be recovered from a variety of plant material and debris, comparatively little work has been done to characterize such strains relative to pathogenic isolates, as evidenced by the isolation of two potentially new species in this manuscript. However, there are several aspects that could be improved to strengthen the overall manuscript.
- Regarding the text, there are many places where the narrative is difficult to follow, is repetitive with previous sections, or seems out of place (e.g., context and justification for experiments sometimes in Methods rather than Results, text appropriate for legends in Results, etc.). Explicitly justifying experiments and bioinformatic analyses, as well as providing context for the results, will help the reader understand the interesting work the authors have done and follow the narrative they have presented.
- The bioinformatics analyses would be strengthened by providing specific detail about parameters used as well as running quality control assessments to estimate assembly completeness (a critical assumption to verify for the comparative genomics section). Below, I have provided specific suggestions and comments:
 - **Line 198.** Need to add reference for these primers.
 - The references for all primer sequences used in this study, their annealing temperatures and the corresponding nucleotide sequence are summarized in Table S1.
- **Line 211-212.** "...indicate possible recombination". This sentence seems more appropriate for the figure 2 legend.
 - The sentence has been transferred to the legend of Figure 2 as follows: Figure 2: Phylogeny of crown gall-associated *Xanthomonas* strains isolated in this study based on the concatenated

sequences of four housekeeping genes, i.e. *atpD*, *efp*, *gyrB*, and *rpoD* using tree-based Maximum Likelihood (A) and network-based NeighborNet (B) analyses. The five *Xanthomonas* strains isolated from weeping fig and *Amaranthus* sp. were divided into two clades where the weeping fig strains were closely related to but still distinct from the *X. translucens* species complex. The *Amaranthus* sp. strain was distinct from all the validly described species with the clade-I of *Xanthomonas*. Further, parallel lines/branches within the nodes of NeighborNet network indicate possible recombination between the strains being connected to each other.

- **Line 223-225.** It would be useful for the reader to add a comment regarding if these genome sizes and GC% are typical for *Xanthomonas* Clade I strains.

- The sentence has been changed as follows: Genome size of the strains was 5,383 kbp (in 152 contigs: FX1); 5,383 kbp (in 89 contigs: FX4), and 5,069 kbp (in 75 contigs: AmX2), with the GC% ranging from 68.8% to 69.8% which is within the typical range of clade I *Xanthomonas* strains.

- **Line 237-244.** Please clarify which of the three methods was used to generate the data shown in Table 2, currently it is not clear. It may also be helpful to spell out the full name of the abbreviations ANI and dDDH the first time they are used. Add ANI and dDDH values for species threshold here for reader comprehension. It would be helpful here to explicitly state the significant finding that AmX2 is below the ANI threshold for species identity with FX1/FX4 – referring to ‘other *Xanthomonas* species’ is ambiguous and could refer to the reference strains used rather than FX1/FX4. Finally, it would be helpful to give the reader an idea of how comprehensive the reference strains included in the ANI analysis were and how many in total were included out of available data – were all clade-I strains used...?

- The following sentence has been added to the text to further clarify the method: In order to estimate DNA similarity indexes between the strains sequenced in this study and all *Xanthomonas* species, ANI values were calculated using three online services JSpeciesWS, ANI calculator, and OrthoANIu where the ultimate ANI value was obtained via combination and averaging of the results generated by the three online services.

- The full name of the abbreviations ANI and dDDH have been spelled out in their first use in the text.

- The following sentence has been added to the text to for readers’ comprehension: These ANI and dDDH values are far below the accepted threshold for prokaryotic species definition, i.e. $\leq 95\%$ and $\leq 70\%$ for ANI and dDDH, respectively, suggesting that the strains FX1, FX4, and AmX2 belong to two new species within the clade-IA of *Xanthomonas*.

- The following sentence has been added to the text to highlight the taxonomic distinction of FX1, FX4, and AmX2: These ANI and dDDH values are far below the accepted threshold for prokaryotic species definition, i.e. $\leq 95\%$ and $\leq 70\%$ for ANI and dDDH, respectively, suggesting that the strains FX1, FX4, and AmX2 belong to two new species within the clade-IA of *Xanthomonas*. While the two strains FX1 and FX4 belong to the same species, the strain AmX2 could be considered as another standalone species both distinct from all validly-described *Xanthomonas* spp. (Figure 3A)..

- The following sentence has been added to the text to clarify how comprehensive the reference strains included in the ANI analysis were and how many in total were included out of available data: According to the MLSA results which confirmed taxonomic position of the strains isolated in this study in clade I of *Xanthomonas* spp., type strains of all clade I *Xanthomonas* species were included in DNA similarity analyses.

- **Line 254.** Missing reference to figure 3A?

- The reference for Figure 3A has been added to the text.

- **Line 258-263.** “...T3SS repertoires...” It would be useful to briefly reintroduce the T3SS, T3E’s, the rationale for looking at the T3SS/T3Es, the implications of lacking a T3SS, and the lack of a T3SS in FX1/FX4/AMX2 before pointing the reader to Figure 3B.

- A short description of the T3SS, T3E’s and TALEs is already existed in the introduction section.
- The following sentences have been added to the text to highlight the implications of lacking a T3SS, and the lack of a T3SS in FX1/FX4/AMX2: It has been shown that the clade I Xanthomonas species possess high genetic diversity in terms of T3SS, T3Es, and TALEs. Genome-informed investigation of Xanthomonas spp. would provide valuable information on the virulence repertoires, pathogenicity mechanisms, and host adaptation of the strains (9).

- **Line 268-273.** “...high query coverage but low sequence similarity...” Is it possible that these are flagellar proteins?

- The following sentences have been added to the text to further clarify the issue: For instance, a sequence similar to hrpB6 (94% query coverage and 48% sequence similarity) was predicted in AmX2, FX1, and FX4. The hrpB6 is a T3SS ATPase in *X. euvesicatoria* (GeneID: 61777304). Furthermore, hrpC2 (originally described in *X. euvesicatoria*; accession number P80150) which belongs to FHIPEP (flagella/HR/invasion proteins export pore) family was detected in all the three strains with 97% query coverage and 32% similarity.

- **Line 277.** I suggest introducing a paragraph break at “As for the....”.

- A paragraph break has been inserted in this position.

- **Line 287-289.** “Circular mapping...” Can the authors clarify if this was sequence read mapping? If so, explicitly stating that mapping sequencing reads against reference genomes provides additional support that sequences appearing in that reference genome are truly absent (or present) in the strain from which reads were generated (rather than simply being unassembled) would be helpful to provide context for this analysis to the reader. That said, while it is clear for some T3Es, it’s very difficult to see that T3SS structural genes are absent based on Figure S1. This figure would be more compelling if it zoomed in on that region, or maybe if read mapping depth vs. average genome mapping depth was provided for these genes from the sequenced strains.

- Please note that we used the FASTA file of the genome sequences obtained in this study for circular mapping. Furthermore, quality assessment of genome assembly and annotation using BUSCO online service confirmed the accuracy and completeness of all three genomes compared to the complete genome sequence of *X. translucens* DSM 18974 (**Line 219-223**). Hence, it could be hypnotized with a considerable confidence that the presence/absence scheme of the genes provided by circular mapping is reliable in the extent of this manuscript. Finally, BRIG software using which the pairwise genome collinearity alignment of the three strains was performed and visualized does not have any zoom in/out option so that the generated circular map could not be zoomed out. In conclusion, we believe that the dataset provided in these analyses are satisfactory enough to shed a light on the genomic contents and virulence-associated repertoires of the Xanthomonas strains isolated from crown gall tissues.

- **Line 306-311.** “One-vs-all collinearity test...” This analysis needs some work, or possibly removal. My interpretation of figure 4B is that the authors have used Mauve to re-order contigs against the strain DSM 18974 reference genome prior to alignment, which should be noted but is absent from this description as well as the methods (**Line 640-641.** “...Mauve software was used....”). Additionally, I

would caution the authors against drawing conclusions about chromosomal structural variation based on re-ordered contigs from draft assemblies. Mauve does not take into account any potentially known spatial relationship data between contigs (i.e., from an assembly graph), only generates a 'best-fit' alignment of the draft to the reference. Fundamentally, alignments like this are similar to reference guided assemblies in that the output for each strain will resemble the reference as much as possible, which is not necessarily the 'ground truth' for a novel strain. While it is entirely possible that any LCB identified which spans multiple contigs in one strain but not others does indeed have that arrangement and the contig break has been introduced by a repetitive sequence insertion, it is also entirely possible that a rearrangement has taken place and those contigs are not adjacent. The teal contig that spans 3000000 on DSM 18974 is an example of this situation, where a contig break is present in FX1 but not the other strains. A second example is the purple contig spanning 2000000 on DSM 18974, which is adjacent to a green contig in FX1 and FX4, but separated from the green contig in FX1 by a contig break – while this alignment looks like FX1 and FX4 have a shared rearrangement relative to DSM 18974, the contig break suggests that FX1 and FX4 may not be colinear in this region. It does not help that the contig breaks are difficult to distinguish from LCB connecting lines in this case. Sequence read mapping, PCR amplification, or long-read sequencing could potentially validate any rearrangements shown here, depending on the size of the gap between contigs. Unfortunately, based on the evidence currently presented, I cannot draw meaningful conclusions regarding collinearity or rearrangements among these strains, only that the new strains seem like relatively complete assemblies relative to DSM 18974. I suggest the authors provide additional evidence for the LCB organization results mentioned in the text, or remove this particular analysis.

- We do agree with the explanations provided by the reviewer, and admit that the sequence data provided in this study are not appropriate to be analyzed with Mauve. The co-linearity analyses which were performed by Mauve software to illustrate locally collinear blocks among the same set of genomes were removed from the manuscript. Figure 4 has been changed accordingly.

- **Line 311-327.** "Three strains sequenced in this study....". Genes shared exclusively between FX1 and FX4 are potentially very interesting given their shared habitat. It would be valuable to indicate how many of these exist, and any potential annotations.

- The text has been changed as follows: Orthologous gene clusters were determined using OrthoVenn online service through four-vs.-four designations of the strains where one of the clade-I strains DSM 18974 and CFBP 4691 were used as reference genome in each analyses (Figure 4B, C). When *X. translucens* strain DSM 18974 was considered as reference, the three strains sequenced in this study shared 611 proteins besides the 2,609 proteins which were shared among all the four strains. The strains AmX2, FX1, and FX4 possessed 35, four and three unique proteins, respectively when DSM 18974 was designated as reference genome. Furthermore, FX1 and FX4 shared 584 proteins with each other while these strains shared respectively 29 and 35 proteins with AmX2 indicating closer genomic repertoires of FX1 and FX4. On the other hand, when *X. theicola* CFBP 4691 was designated as reference genome the three strains sequenced in this study showed 811 shared proteins along with 2,416 proteins which were shared within all the four genomes. The strains AmX2, FX1, and FX4 possessed 37, three and three unique proteins respectively, when CFBP 4691 was considered as reference genome (Figure 4B, C).

- **Line 341-344.** "prophage sequences.... that were missing...". It's worthwhile to note that because these are draft genomes, it is possible that more phage sequences (which are often repetitive) are present in these strains but were not identified in the present study due to unresolved repetitive sequence.

- The following sentence has been added to the text: However, it should be noted that due to the draft nature of the genomes, it is possible that more phage sequences are present in these strains but were not identified in the present study due to unresolved repetitive sequence.

- **Line 345-360.** “Visualization of genomic islands....” This would benefit from additional explanation, both about genomic islands as well as the results. Did any of these islands contain interesting content, or is there evidence of horizontal gene transfer, or is there an island shared among strains....? Currently this feels tacked-on with no context for the reader.

- The following explanation has been added to the text to clarify the status of genomic islands in the strains sequenced in this study: Visualization of genomic islands in the whole genome sequences of the three *Xanthomonas* sp. strains obtained in this study against the genomes of two reference strains *X. theicola* CFBP 4691 and *X. translucens* pv. *translucens* DSM 18974 is shown in Figure S2. Distribution pattern of the main genomic islands in all the three strains sequenced in this study remained constant regardless of the reference genome used for prediction. When the strain CFBP 4691 was considered as reference genome, two dominant genomic islands were predicted in FX1 positioning between 0.9M-1.0M and 2.5M-2.6M. The two islands were predicted when DSM 18974 was used as reference although the position of the islands was slightly different (1.5M-1.6M and 2.8M-2.9M). Similarly, a genomic island was constantly predicted in FX4 when both the reference genomes were used, while position of the island has changed from 2.1M-2.2M to 2.9M-3.0M when CFBP 4691 and DSM 18974 were used as reference genome, respectively (Figure S2). The same pattern was observed in the strain AmX2 where a dominant genomic island was predicted on 1.6M-1.7 and 2.3M-2.4M when CFBP 4691 and DSM 18974 were used as reference genome, respectively. These results indicate different patterns of genomic islands distribution among the strains sequenced in this study despite the fact that they belong to the same species, i.e. FX1 and FX4 (Figure S2).

- **Line 364-372.** “genomic repertoires of five...strains...”. I recommend rephrasing this section to be clear that five strains were isolated and phenotypically characterized, but only three were sequenced and compared.

- The text has been changed as follows: Five *Xanthomonas* sp. strains have accidentally been isolated from crown gall tissues along with several *Agrobacterium* strains, while the latter organisms were the main causal agent of the disease. In this study, we employed a series of biochemical tests, pathogenicity assays and MLSA-based phylogenetic analyses on five crown gall-associated *Xanthomonas* strains isolated from weeping fig and *Amaranthus* sp. plants in Iran. Subsequently, three representative strains, i.e. FX1, FX4 and Amx2 were subjected to genome sequencing and comparative genomics to elucidate their biological features, taxonomic position and genomic repertoires. The strains investigated in this study induced hypersensitive reaction (HR) on geranium, melon, squash, tobacco and tomato leaves while they were non-pathogenic on their host of isolation. MLSA-based phylogenetic analyses accomplished with whole genome sequence-based ANI/dDDH calculations suggested that the *Xanthomonas* strains isolated from crown gall tissues could be considered as two novel species within the clade-I of *Xanthomonas*.

- **Lines 396-404.** Some of these sentences seem out of order or repetitive.

- The text has been re-phrased as follows: A physiologically active crown gall produces specific nutrients (opines) utilizable by microorganisms providing a rich niche for plant-associated bacteria to colonize, grow, and form complex ecological interactions (19, 20; 21; 22). The microbial communities involved in opine ecology is not limited to the *Agrobacterium* members.

Complex bacterial communities of many species co-exist in crown gall tissues (23). During the past few decades, various bacterial species, i.e. *Pseudomonas* spp. and coryneform bacteria were isolated from crown gall tissues possessing the capability of competing for opines as growth substrates (19).

- **Line 331-337.** "It could also be attributed to...". Were opine catabolism genes identified in these strains, maybe homologs of *ooxAB*? Can they grow on opines as a sole carbon source? I recommend rephrasing this sentence so it is clear that these strains if these strains have known opine catabolism genes or not.

- The following sentences have been added to the text: BLAST-based in-silico analyses using the sequences of *ooxA*, *ooxB*, and *ooxAB* genes against the three genome sequences obtained in this study revealed the presence of these genes in the three *Xanthomonas* strains. As detailed in Table S2B, sequences of DNA fragments with 66-98% query coverage and 25-27% sequence similarity to *ooxA*, *ooxB*, and *ooxAB* genes were detected in the genome of all three strains. These evidences suggest the potential acquisition of opine catabolism genes by gall-associated xanthomonads which needs further in-depth investigations.

- **Line 478-482.** "non-pathogenic *Pseudomonas* strains..." these strains also typically fail to induce HR on tobacco, unlike the *Xanthomonas* strains in this study - a notable difference.

- The following sentence has been added to the text: These non-pathogenic *Pseudomonas* strains also typically fail to induce HR on tobacco, unlike the *Xanthomonas* strains isolated in this study.

- **Lines 482-494.** "An in-depth comparative analysis....several lines of evidence...". This is very vague. This section would be much stronger if the authors explicitly discussed the lines of evidence and precise insight gained.

- The following sentences have been added to the text: Subsystem analyses using RAST service highlighted differences in the genetic contents of clade-I xanthomonads especially when it comes to the differences between the cereal pathogen *X. translucens* and the strains isolated in this study. For instance, the number of subsystems that are responsible for biotin biosynthesis, invasion and intracellular resistance, siderophore production, protein biosynthesis, nitrogen metabolism, central aromatic intermediates metabolism, sulfur metabolism, methylglyoxal metabolism, potassium metabolism, maltose and maltodextrin utilization, respiration and the miscellaneous subsystems in the strains FX1 and FX4 was higher than those in the members of their neighbor clade *X. translucens*. Furthermore, prophage sequences were missing in the strains sequenced in this study, but since these are draft genomes, it is possible that more phage sequences (which are often repetitive) are present in these strains but were not identified in the present study due to unresolved repetitive sequence.

- **Lines 538-540.** I recommend moving the HR information to the Plant Inoculation Assay section beginning at line 538.

- Done.

- **Line 565.** "...weight of each leaf sample was adjusted to 2 g". Could this influence the results in Figure 1 for comparisons between hosts? For example, by including far more inoculated leaves from one species by virtue of lower leaf mass and thus more CFUs.

- We do agree with the reviewer's opinion that using similar weight of all plant species could possibly influence the final data. However, if one wants to use different weights of leaf samples in different plant species, the resulting data would be incomparable, and it would be almost

impossible to summarize and present the data in one graph. To create uniform experimental conditions in this experiment, we took the same weight from the leaves.

- **Lines 559-561.** “Further, in order to...”. This is an example of information that would help the reader follow the narrative in the Results section (**Lines 178-180**), and is unnecessarily duplicated here

- The following sentences have been moved to the Results section: Furthermore, in order to provide a precise insight into the biology of the strains isolated in this study, in planta growth of the strains AmX2 and FX1 was investigated on 12 plant species, i.e. Amaranthus sp., barley, black nightshade (*Solanum nigrum*), red nightshade (*Solanum villosum*), common bean, common zinnia (*Zinnia elegans*), maize, melon, spinach (*Spinacia oleracea* L. cv. hudson), squash, watermelon, and wheat under greenhouse conditions. Population size of the inoculated bacterial strains were counted 7, 14, 21, and 28 dpi and presented in log CFU/g of the leaf tissue (Figure 1I, J).

- **Line 599-603.** Please provide more information regarding genome assembly, specifically any relevant non-default parameters used for bbduk, trimmomatic, SPAdes (e.g., --careful, --isolate), which SPAdes pipeline was run, if any steps were taken to correct indels present in the assembly, etc. I also strongly recommend that the authors run at least one quality control analysis to assess the completeness and potential contamination in these genome assemblies. CheckM, BUSCO, or QUAST (using a high-quality clade I *Xanthomonas* reference genome) would be appropriate. A high completeness score with these methods would lend additional support to every genomics analysis in this manuscript, especially the absence of a T3SS and T3Es as a true feature of these genomes rather than potentially unassembled sequence.

- Quantitative assessment of genome assembly and annotation completeness was evaluated using BUSCO online software.

- The following sentences have been added to the text: The read quality was assessed with FastQC (Galaxy version 0.72+galaxy1; <http://www.bioinformatics.babraham.ac.uk/projects/fastqc/>), while quantitative assessment of genome assembly and annotation completeness was evaluated using BUSCO online software (46; 47).

- **Line 343-344.** Were results from PlasmidFinder presented in this manuscript? I could not find them.

- The following sentence is the result of PlasmidFinder analyses: No integrative plasmid (episome) was detected using the PlasmidFinder online service in the genome of strains in this study.

- **Lines 660-662, and 676-677.** Delete duplicated section and add BioProject information, SRA information if submitted, etc.

- Done.

- BioProject data of the strains have been added to the text as follows: Whole genome sequences of the three strains were also deposited into the NCBI GenBank database under accession numbers for AmX2: JAFIWC000000000 (BioProject: PRJNA700118); FX1: JAFJNT000000000 (BioProject: PRJNA700116); and FX4: JAFIWB000000000 (BioProject: PRJNA700117).

- **Lines 917-918.** Please indicate what measurement is represented by the error bars in Figure 1.

- The following sentence has been added to the legend of Figure 1: Error bars indicate standard deviation.

- **Line 944-953** (Figure 4C). What do the question marks (“?”) indicate on the OrthoVenn diagram?

- Figure 4 has been redrawn completely.

- Table 1. I recommend adding an additional score to clarify that the *Xanthomonas* sp. induced HR on Melon, Squash, etc. (e.g., 'HR', or '-*' instead of '+'). The current presentation suggests that these strains are pathogenic on those hosts, whereas the evidence presented in lines 154-181 suggests that these strains are not pathogenic on the evaluated hosts.

- Table 1 has been corrected accordingly.

- Table 1, top row, change "Common been" to "Common bean".

- Corrected.

- Figure 1, This figure would benefit from images of control treatments for hypersensitivity experiments e.g., sterile buffer, a *Xanthomonas* strain that does not induce HR on these hosts, and a *Xanthomonas* strain known to induce HR on these hosts. Additionally, providing a CFU estimate for the bacteria suspensions would help reproducibility. I agree with the previous reviewer #3 that 1G-1J can be presented more clearly. If I understand the author's intention correctly, these plots should show that there is no meaningful bacterial population growth over time in these hosts, providing evidence that these strains are not pathogenic on these hosts, which is currently very difficult to see. For example, this data could be represented on two plots, one per strain, by clustering all four time points per host together on one axis, with different colors or patterns differentiating the time points. If a strain-vs-strain comparison is desired, one plot could be made using the same approach. Either of these presentations would focus on the time course aspect of this experiment and significantly improve the reader's comprehension, while still allowing for a host-vs-host comparison.

- Figure 1 has thoroughly been revised and all requested items have been added.

- Table S5. I recommend providing contig IDs, or gene IDs instead of start and end codons, for these proteins for genomes with more than one contig.

- The gene IDs have been added to the table.

- Table S6. Providing contig IDs would be useful here as well.

- The gene IDs have been added to the table.

Reviewer comments:

Reviewer #2 (Comments for the Author):

- In the manuscript entitled "Phenotypic and Molecular-Phylogenetic Analyses Revealed Distinct Features of Crown Gall-Associated *Xanthomonas* Strains", Mafakheri et al. described that they isolated several bacterial strains belonging to the genus *Xanthomonas* and found that these strains are associated with crown gall disease of plants. The authors used MLSA method to classify the phylogenetic relationships of the strains against the *Xanthomonas* spp, and revealed that FX1-FX4 are close relatives of *X. translucens*, whereas Amx2 is in a special clade. Furthermore, they compared the genomes and gene contents of these strains with the representative bacterial species of *Xanthomonas*, which showed that the crown gall-associated strains have different gene compositions in virulence-associated genes.
- Although the data reported in the manuscript is interesting to the readers of plant pathology, especially those study crown gall diseases, the work carried out preliminary. The first complete genome sequence and comparative genomic analysis of *Xanthomonas* have been reported for nearly 20 years, and there is numerous sequence data of the genus deposited in the public databases. Bacterial strains isolated in this work are very special because the role of *Xanthomonas* spp. in crown gall disease was overlooked.

However, descriptive and comparative analyses failed to address some of the important questions, such as:

- Is *Xanthomonas* spp. alone or the cooperation of these strains with agrobacterium caused the crown gall?
 - As indicated in the manuscript, the *Xanthomonas* strains isolated in this study did not induce any symptoms on their host of isolation nor on the other evaluated test plants except for HR which was observed on some annual crops. On the other hand, the accompanying *Agrobacterium* strains isolated from the same gall tissues caused crown gall symptoms on the test plants either in presence or absence of the *Xanthomonas* strains. Hence, the gall formation is a characteristic of *Agrobacterium* strains that is not affected by *Xanthomonas* strains. This issue is discussed in the manuscript. Furthermore, we have evaluated the opine-catabolic operon named *ooxAB* in the *Xanthomonas* strains isolated from gall tissue. The results are described in **lines 431-436**.

- How about the gene function in regulating virulence of development of the crown gall, as comparative genomic analysis predicted?
 - As explained in the previous comment, the *Xanthomonas* strains isolated in this study did not induce any symptoms on their host of isolation nor on the other evaluated test plants. Furthermore, they did not affect the gall formation function of their accompanying *Agrobacterium* strains. We do not claim any correlation with the genomic content of the *Xanthomonas* strains and formation of crown gall in the host plants. Comparative genomics of the *Xanthomonas* strains was performed to shed a light on the genetic contents of the strains in comparison with the other *Xanthomonas* species but not necessarily to find a correlation between their genomic content and the gall formation. Furthermore, we have evaluated the octopine-catabolic operon named *ooxAB* in the *Xanthomonas* strains isolated from gall tissue. Evidences of the presence of these genes in the gall-associated *Xanthomonas* strains have been described in **lines 431-436**.

Reviewer #3 (Comments for the Author):

- *Reviewer's comment:* Authors have isolated *Xanthomonas* strains from crown-gall samples, conducted Koch's postulates, and analyzed their genomes. Given the previous studies indicating that absence of T3SS and effectors in xanthomonads belong to clade I isolated from wide variety of hosts, authors need to justify how this study advances our understanding of these non-pathogenic xanthomonads. How are your findings relevant to plant-microbe interaction studies? Authors can improve this section rather than repeating what is already stated in the abstract. -
- *Author's response:* As stated in the "Importance" section of the manuscript, the microclimate of crown gall and their accompanying microorganisms has rarely been studied for the microbial diversity and population dynamics of gall-associated bacteria. In this study, we highlight the phenotypic characteristics and genomic features of gall associated xanthomonads. Our analyses not only revealed that biological behaviors and compatibility of *Xanthomonas* strains in the presence of their agrobacterial counterpart, they also shed light on the phylogenetic position, taxonomic status and virulence-associated genomic feature of the strains.
- *Reviewer's reply:* What do you mean by compatibility? Have authors really addressed biological behavior of the *Xanthomonads* in presence of agrobacterial counterparts? Have they looked at population dynamic of gall associated xanthomonads in presence of *Agrobacterium* on Amaranth? The manuscript starts with crown gall associated *Xanthomonads*, but then focus suddenly shifts to the clade I xanthomonads and diversity within clade I. It seems that somehow overall question of ecology of these xanthomonads in presence of *Agrobacterium* is not addressed.

- As we stated in the manuscript a couple of greenhouse experiments highlighted the biological properties of the *Xanthomonas* strains. First, we have evaluated the pathogenicity and host range of the strains not only on their host of isolation but also on some other annual crops using the *Xanthomonas* strains alone or in combination with their accompanying *Agrobacterium* strains. Second, we have evaluated the epiphytic growth and survival of the *Xanthomonas* strains on a series of annual crops which provides an insight into the extent of the possibility of their survival on the plant species other than their host of isolation.
- Furthermore, we have added a number of new analyses i.e. the opine catabolism genes in the *Xanthomonas* strains. We have evaluated the octopine-catabolic operon named *ooxAB* in the *Xanthomonas* strains isolated from gall tissue. The results are described in **lines 431-436**.

- *Reviewer's comment:* Fig 1D-F- if you observed water-soaking phenotype, it is clearly not hypersensitive response! Authors have this completely wrong interpreted. Why was tomato not included in the population? Why was wheat phenotype not shown? Based on phylogenetic placement of these strains, one would expect to have these strains tested on hosts on which closely related strains are pathogens, eg. Grains, tea, etc.

- *Author's response:* Fig 1D-F (**Lines 165-171**): The sentences have been changed as follows: In the leaf infiltration method, hypersensitivity reaction i.e. necrotic areas with a blackish-brown appearance was observed in the site of inoculation on squash (Figure 1D) melon (Figure 1E), and tomato (Figure 1F) as well as tobacco and geranium leaves 36-48 hpi. - Please note that we have inoculated wheat plants with both spray inoculation and injection techniques. Since we did not observe any symptoms on wheat plants inoculated with the bacterial strains, it does not make sense to include a picture of asymptomatic wheat plant.

- *Reviewer's reply:* If there is hypersensitivity reaction and based on your observation of T3SS presence/absence, do you think these strains get recognized by different plant species? If not, how are these strains able to maintain their populations on the non-hosts?

- Please note that the survival of plant pathogenic bacteria on non-host plant species is not necessarily controlled by their pathogenicity repertoires. Rather, there are several examples in the literature where it has been shown that both non-pathogenic and pathogenic variants of a given bacterial pathogen have almost equal epiphytic survival on non-host plant species. To see one of those examples please visit <https://bsppjournals.onlinelibrary.wiley.com/doi/full/10.1111/ppa.12730>

- *Reviewer's comment:* Any other better way to represent fig 1G to J? It is hard to compare populations over time when plotted as separate graphs. Why not arrange growth curves by host? It is interesting to see populations on certain hosts do not change over 28-day period. Why? Answering above questions might be useful to interpret about non-pathogenic nature of these strains.

- *Author's response:* It could be possible to present the population dynamics on each plant species as a separate graph. However, in that case it would be very hard to compare different plant species with one another in the course of time.

- *Reviewer's reply:* Refer to my above comment again. What is the rationale to compare populations among different plant species at a given sampling time? Isn't it easier to interpret on strain's ability to sustain populations on a given host by following growth curve/survival curve over time? It is interesting that wheat showed higher populations, yet it was asymptomatic among all hosts tested.

- Figure 1 has thoroughly been revised and all requested items have been added.

- *Reviewer's comment:* There are several analyzes conducted in the comparative genomics section. For example, bacteriocins, phage, genomic islands. Authors might want to state the rationale for conducting

these analyses. What is your hypothesis? In absence of a hypothesis, this laundry list of analyzes make no sense.

- *Author's response:* These analyses altogether provide a comprehensive insight into the genome arrangement, virulence repertoires, and potential biological capabilities of the new xanthomonad clades. Since these strains represent a new set of taxa in terms of phylogenetic position and taxonomic status such analyses seem to help better understanding of the genetic diversity within the clade I of xanthomonads.

- *Reviewer's reply:* Authors have convinced us on the diversity within clade I. However, how does fig 4A data help you to infer on ecology of Xanthomonads co-occurring with Agrobacterium?

- Please note that comparative genomics analyses using the data provided by RAST and SEED-Viewer databases highlighted differences in metabolic networks and subsystem profiles between the three strains sequenced in this study and those of the clade-I xanthomonads, as well as the reference strain *X. oryzae* pv. *oryzae* PXO99A. Results of these analyses will help the readers to find metabolic characteristics that differentiate the strains isolated in this study and those of plant pathogenic *Xanthomonas* spp.

- Overall manuscript writing needs significant amount of work. I miss overall rationale behind this study. I see several pieces that authors are trying to put together, such as comparative genomics, pathogenicity tests, new species proposition etc. But what does it mean to simultaneously isolate these strains from crown gall samples? Is studying genomes to explain ecology of these strains one of the goals?

- *Author's response:* This is a preliminary study to identify, characterize and describe the new atypical xanthomonad strains isolated from crown gall tissues of plants simultaneously infected with Agrobacterium strains. The phenotypic and genotypic data provided in this study along with the whole genome sequence resources will further pave the way of research on the clade I of xanthomonads. This study follows our previous projects i.e. <https://journals.asm.org/doi/full/10.1128/AEM.01518-19>, <https://apsjournals.apsnet.org/doi/10.1094/PHYTO-11-19-0428-R>, and <https://journals.asm.org/doi/full/10.1128/AEM.01518-19> on the clade I of xanthomonads. Further genomic investigations are undergoing to shed a light on the evolutionary relationships of the strains isolated in this study with those infecting small grain cereals.

- *Reviewer's reply:* Authors have been engaged with significant research on clade I xanthomonads. Overall, the manuscript falls short of connecting the dots. If exploring clade I diversity is the goal, authors have convincing data. But if characterizing xanthomonads simultaneously infecting along with Agrobacterium, there is more to infer from the analyses already conducted in this manuscript.

- The ultimate goal of this manuscript is to provide preliminary data on the biological characteristics and phylogenetic status of the gall-associated *Xanthomonas* strains and their taxonomic relationships with the clade I xanthomonads.

December 8, 2021

Dr. Ebrahim Osdaghi
University of Tehran
Department of Plant Protection
Department of Plant Protection
University of Tehran
Tehran
Iran

Re: Spectrum00577-21R2 (Phenotypic and Molecular-Phylogenetic Analyses Revealed Distinct Features of Crown Gall-Associated Xanthomonas Strains)

Dear Dr. Ebrahim Osdaghi:

Your manuscript has been accepted, and I am forwarding it to the ASM Journals Department for publication. You will be notified when your proofs are ready to be viewed.

Sincerely,

Lindsey Burbank
Editor, Microbiology Spectrum
